# Asymmetry and redundancy of STAT5 paralogs across CD8+ T cell differentiation states
Svetlana Ristin[1,2,5], Molly Dalzell[1,2,5], Christopher Armstrong [1], Nisa Ilsin[1], Antonio M. Fontanella [1], Luis Nivelo[1,2], Lothar Hennighausen[3], John J. O'Shea[4,6] & Alejandro V. Villarino [1,2,6] ✉

Fostering STAT5 signaling is key to immunotherapies that leverage CD8+ T cell biology. Using mouse models, we demonstrate that the two mammalian STAT5 paralogs, STAT5A and STAT5B, are at once redundant and functionally distinct in CD8+ T cells. Thus, they are *asymmetric paralogs*, exhibiting both widespread homology at molecular level and functional asymmetry at cellular level, with STAT5B emerging as dominant. For mechanisms, we determined that STAT5B is twice as abundant, accounting for two-thirds of the total STAT5 pool, and present evidence that it also has distinct, paralog-specific properties. We also defined cytokine- and cell state-restricted STAT5B functions and devised a core signature that spotlights key downstream properties and serves as bioinformatic probe. Together, these studies affirm the centrality of STAT5 in CD8+ T cells, reveal common and circumscribed activities for STAT5A and STAT5B, and present a unifying model that foregrounds both redundancy and asymmetry.

STAT5 is a signal-dependent transcription factor that mediates key cellular pathways in immune cells, ranging from universal, pan-lineage processes like glucose metabolism and apoptosis, to specialized, lineage-specific processes like cytokine production and cytotoxicity[1–6]. STAT5 has been most extensively studied in lymphocytes, especially T cells, where it exerts numerous pro- and anti-inflammatory functions. Examples of the former include its ability to promote CD8+ and NK cell cytotoxicity[7–10], and its ability to promote IFN-γ production across the lymphoid compartment[2,3,10–12]. Examples of the latter include its ability to marshal CD4+ regulatory T cells (Treg) and to suppress Th17- and follicular helper-type (Tfh) T cell responses[13–21]. Genetic studies illustrate both the importance of STAT5 and its dual pro- and anti-inflammatory nature. Complete germline ablation in mice results in severe lymphopenia and anemia leading to perinatal death, and the few animals that survive beyond one month eventually develop lethal, systematic autoimmunity[22–25]. Partial germline STAT5 deficiency also manifests a paradoxical combination of lymphopenia and autoimmunity, only less severe, and, in mice, pathology is localized mainly to kidneys[26–30].

STAT5 is unique among STAT family members in that it is a collective of two proteins, STAT5A and STAT5B. Thus, the mammalian genome has 4 total *STAT5* alleles encoded by two adjacent loci, *STAT5A* and *STAT5B*, resulting from an evolutionarily recent duplication event[31,32]. Given this recent divergence, STAT5A and STAT5B are ~95% homologous at the protein level, which has led to persistent questions about whether they are redundant or functionally distinct. Arguments are compelling on both sides. First, genome-wide distribution studies have established that they mostly co-localize and, in turn, regulate many of the same genes[7,33,34]. Second, in a qualitative sense, STAT5A and STAT5B deficiencies impact the immune system in similar ways. For instance, both result in diminished NK, Treg, and CD8+ T cells[7,30,33,35]. However, they are not similar in a quantitative sense. Immunological phenotypes are more pronounced in mice lacking *Stat5b* than those lacking *Stat5a*, and most dramatic in mice lacking both *Stat5a* and *Stat5b*, which suggests cooperation and/or STAT5B-specific functions. For instance, *Stat5b*-deficient mice have far greater reductions in steady state lymphocytes, particularly Innate Lymphoid Cells (ILCs), and far greater defects in cytokine-driven T cell responses[7,22,30,33,35–37].

In humans, germline loss-of-function *STAT5B* mutations result in severe immunological dysfunction, including sharply reduced and/or defective NK, Treg, and CD8+ T cell compartments. Analogous *STAT5A* mutations have yet to be reported[26–29]. Also, somatic mutations of *STAT5B*

[1]Department of Microbiology and Immunology, Miller School of Medicine, University of Miami, Miami, FL, USA. [2]Sylvester Comprehensive Cancer Center, University of Miami, Miami, FL, USA. [3]National Institute of Diabetes, Digestive and Kidney Diseases, National Institutes of Health, Bethesda, MD, USA. [4]Lymphocyte Cell Biology Section, Molecular Immunology and Inflammation Branch, National Institute of Arthritis, Musculoskeletal and Skin Diseases, National Institutes of Health, Bethesda, MD, USA. [5]These authors contributed equally: Svetlana Ristin, Molly Dalzell.[6]These authors jointly supervised this work John J. O'Shea Alejandro V. Villarino. ✉e-mail: alejandro.villarino@miami.edu

are far more common in T cell cancers than *STAT5A* mutations and, when tested head-to-head, appear more pathogenic[38–40]. Thus, functional asymmetry between STAT5 paralogs factors across human diseases and, as in mice, STAT5B appears dominant over STAT5A. However, it is important to note that, while less potent, gain-of-function STAT5A mutants are still capable of invoking T cell malignancy[9,38,40]. Furthermore, *Stat5a* deficiency is more impactful than *Stat5b* deficiency in some cell types, including mammary epithelium[41–44], hematopoietic stem cells[45], and B cells[46], and both STAT5A and STAT5B are required for optimal tetramerization and elaboration of downstream transcriptional programs[7,47,48]. Thus, while it may not substitute for STAT5B, STAT5A does matter for overall STAT5 function and likely becomes more relevant in settings where STAT5 activity spikes.

STAT5 is activated downstream of many cytokines and growth factors, but, in lymphocytes, is most associated with cytokines operating through the common $\gamma$ chain ($\gamma c$) receptor and the Janus kinase, JAK3[2,49]. Members of this family include IL-2, IL-4, IL-7, IL-9, IL-15, and IL-21. Each employs a dedicated co-receptor subunit that pairs with $\gamma c$ to enable both ligand-specificity and a shared ability to activate STAT5. Downstream of $\gamma c$ and JAK3, STAT5 is required for development and homeostasis of multiple lymphoid lineages, most notably T and NK cells[5,25,50], and directly instructs key genes involved with lymphocyte differentiation, including *IgK* in B cells[51,52], *RUNX3*, a transcription factor necessary for bifurcation of the CD4$^+$ and CD8$^+$ T cell lineages[50], and FOXP3, the "master" transcription factor of CD4$^+$ regulatory T cells (Treg)[15,20,21]. Another important feature is its ability to drive terminal differentiation, the process by which lymphocytes make irreversible fate decisions that enable effector functions. This capacity is clearly evident in CD8$^+$ T cells, where STAT5 promotes cytotoxicity and memory formation/maintenance while subverting "exhaustion", a dysfunctional state which curtails the effector program[8,9].

The central role of STAT5 in lymphocyte biology has motivated research on upstream cytokines as therapeutic agents. In fact, recombinant IL-2 was the first immuno-therapy[53] and numerous trials are ongoing for this and other STAT5-activating cytokines in settings of cancer and autoimmune disease, including recent efforts to: (1) engineer upstream cytokines for "better" or lineage-selective STAT5 activities, (2) combine upstream cytokines with "checkpoint" inhibitors, and (3) incorporate STAT5 signaling payloads in Chimeric Antigen Receptor T cell constructs[54–58]. Using a genetic model, we addressed the evergreen question of whether STAT5A and STAT5B are redundant or functionally distinct, focusing on CD8$^+$ T cells, principal mediators of many immunotherapies. We conclude that they are *asymmetric paralogs*, exhibiting both functional homology at the molecular level and asymmetry at the cellular level. We also identify cytokine-, lineage-, and state-restricted STAT5 functions, and a core STAT5 signature that reflects emblematic STAT5-regulated pathways, such as carbon metabolism and apoptosis. Together, these findings affirm the centrality of STAT5 in CD8$^+$ T cells, reveal context-specific STAT5 activities, and establish a unifying model for functional disparity between STAT5 paralogs.

## Results

### STAT5 depletion markedly impacts the CD8$^+$ T cell compartment

To study the relationship between STAT5 paralogs, we generated a series of germline knockout (KO) mice with diminishing STAT5 alleles, starting with four and dwindling to one (Fig. 1A)[30,35,42]. Hereafter, we refer to each strain according to the alleles that were deleted, save for 4 allele mice, which are "wild type" (WT; *Stat5a*$^{+/+}$ *Stat5b*$^{+/+}$). Thus, AAB mice retain one allele of STAT5B (*Stat5a*$^{-/-}$ *Stat5b*$^{+/-}$), while BBA mice retain one allele of STAT5A (*Stat5a*$^{+/-}$ *Stat5b*$^{-/-}$). We previously used this approach to study CD4$^+$ helper T cells and found that STAT5 deficiency strongly impacted both effector and regulatory functions, with STAT5B emerging as dominant[30]. Importantly, we used the term "dominant" (as we do now) to indicate that STAT5B has a greater relative impact and not as used in genetics, where it refers to a "dominant" allele that overrides effects of a "recessive" allele. During the course of those studies, we also noted that, unlike CD4$^+$ T cells,

CD8$^+$ T cells were depleted in spleens, lymph nodes, and bone marrow[30], resulting in sharply skewed CD4/CD8 ratios (Fig. 1B–E). This phenotype was more pronounced in strains lacking STAT5B than those lacking STAT5A and most dramatic in those with only one STAT5A allele (BBA), suggesting a gene dose effect. Thus, CD8$^+$ T cells appear acutely sensitive to reductions in STAT5 and more dependent on STAT5B than STAT5A.

To determine if the observed phenotypes are T cell intrinsic, we generated mixed chimeras using bone marrow from WT and AAB or BBA donors. A 1:5 ratio of WT to KO donors was used because we expected that WT donors would have a strong competitive advantage. Indeed, despite this consideration, splenic CD8$^+$ T cell ratios were strongly skewed towards WT donors, with the WT/BBA mix trending to greater shifts (Fig. 2A–C). By contrast, splenic CD4$^+$ T cells had donor ratios closer to 1:1, suggesting that they are less reliant on STAT5. To complement these studies, we also analyzed mice lacking both STAT5A and STAT5B selectively in T cells and found that the CD8 compartment was conspicuously depleted in primary lymphoid organs, resulting in sharply skewed CD4/CD8 ratios (Fig. 2D). The same was not true of CD4$^+$ T cells, which were largely unchanged (Fig. S1A, B). Together, these data affirm that CD8$^+$ T cells are more impacted by STAT5 deficiency than CD4$^+$ T cells and, crucially, that the CD4/CD8 skewing seen in our STAT5 allele mice likely reflects T cell intrinsic functions. However, it remains true that STAT5 signaling is globally depressed and that several (if not all) of these mouse lines have some level of chronic, baseline inflammation[30], so we must also acknowledge that cell-extrinsic "knock-on" effects likely contribute to the observed CD8$^+$ T cell phenotypes.

### STAT5 paralog asymmetry in CD8$^+$ T cells

Given sharp reductions in CD8$^+$ T cell counts, we next assessed the impact of STAT5 paralog deficiency on effector and memory differentiation. Using well-described surface markers, we determined that STAT5 deficiency leads to accumulation of both effector and memory CD8$^+$ T cells, at the expense of naive CD8$^+$ T cells (Fig. 3A, B). As with cell counts, the effector/memory phenotype was most pronounced in mice lacking STAT5B, and plainly evident in mice lacking STAT5 selectively in T cells (Figs. 3A–C and S2A–C). Therefore, STAT5 signaling not only enables accumulation of CD8$^+$ T cells in lymphoid organs, but also keeps them in a quiescent, naive state. STAT5A and STAT5B both participate but asymmetrically, with STAT5B emerging as dominant.

STAT5 controls expression of key elements in the CD8$^+$ T cell effector program, including granzymes and cytokine receptors[7,33]. To determine if these are compromised in the absence of STAT5A and/or STAT5B, we purified naive CD8$^+$ T cells from WT, AAB, or BBA mice, activated them with agonist anti-T Cell Receptor (TCR) and anti-CD28 antibodies in the presence of IL-2, a potent STAT5 stimulus, then measured effector proteins by flow cytometry. Expression of Granzyme B and IL-2R$\alpha$ was strikingly reduced in both STAT5A- and STAT5B-deficient cells relative to WT controls while expression of Granzyme A was strikingly increased (Fig. 3D). In all cases, there was a trend towards greater effect in STAT5B-deficient cells which failed to reach statistical significance (Fig. 3D). Thus, while prone to effector differentiation, STAT5-deficient CD8$^+$ T cells cannot elaborate key elements of the cytotoxic program and, in this capacity, STAT5A and STAT5B appear similarly relevant.

STAT5B accounts for approximately two-thirds of total STAT5 in CD4$^+$ T cells[30]. Thus, we next asked if asymmetric expression is also evident in CD8$^+$ T cells. First, we compared total STAT5 protein levels (STAT5A + STAT5B) in CD44$^{low}$ naive and CD44$^{high}$ effector/memory CD8$^+$ T cells from WT, AAB, and BBA mice. As with CD4$^+$ T cells, we found that STAT5B deficiency clearly had greater impact and, in turn, calculated that it accounts for about 60% of the total STAT5 pool in naive and effector/memory CD8$^+$ T cells (Fig. 3E). Similar disparity was evident at transcriptional level where, again, *Stat5b* accounts for 60–70% of total *Stat5* mRNA (Fig. 3F) and holds true whether transcriptomes were generated in-house (detailed below) or mined from public databases (Figs. 3F and S3A–C), and whether sourced from mouse or human T cells

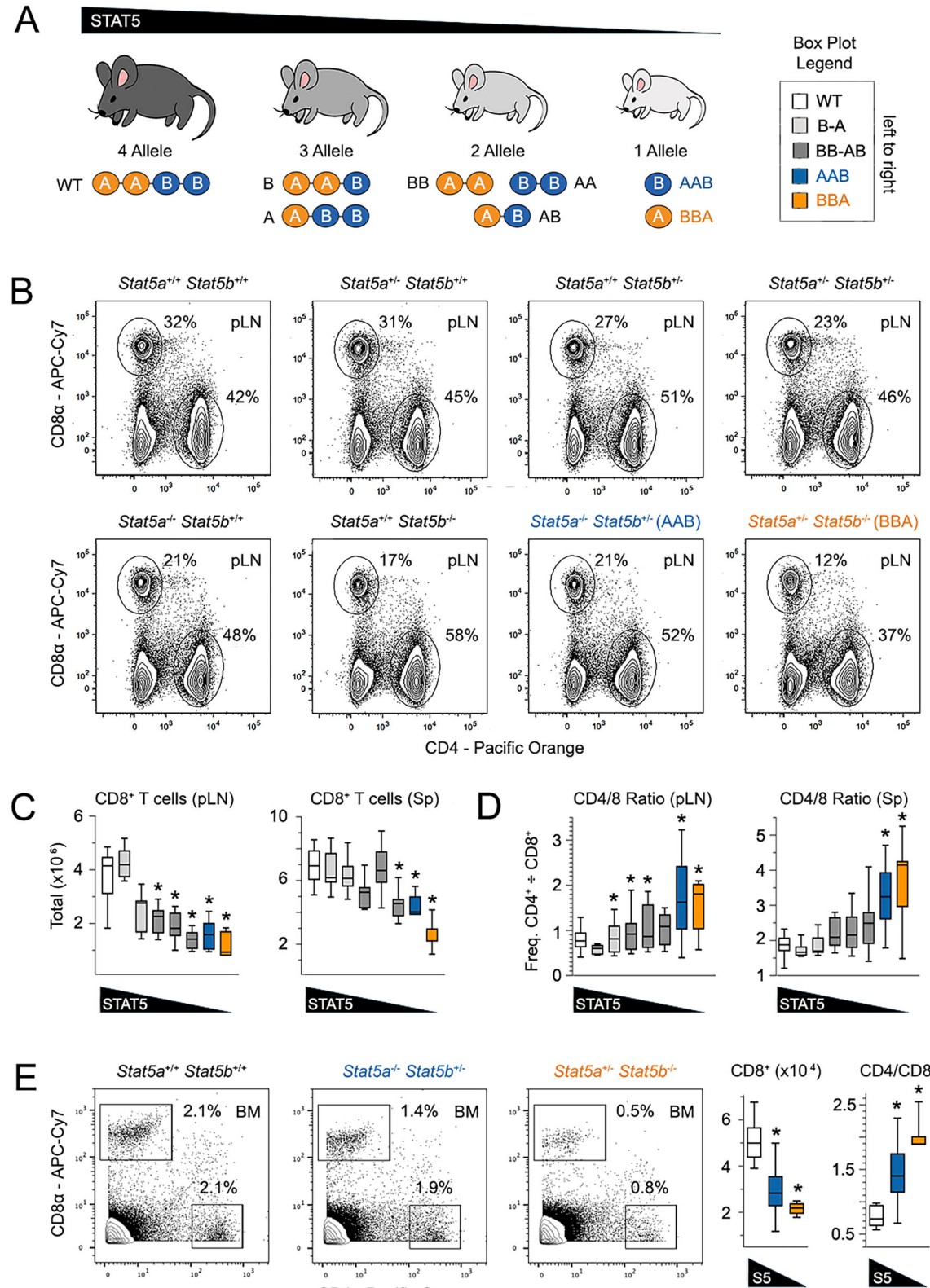

**Fig. 1 | STAT5 deficiency markedly impacts the CD8+ T cell compartment.**
**A** Cartoon depicts mouse models used in this study. The 8 genotypes are grouped based on total STAT5 alleles, ranging from 4 to 1, and named according to alleles that are deleted. AAB ($Stat5a^{-/-}$ $Stat5b^{+/-}$) and BBA ($Stat5a^{+/-}$ $Stat5b^{-/-}$) mice are always colored blue and orange, respectively. Mouse line drawing originally published by the Public Library of Science and used here in accordance with the Wikimedia Creative Commons Attribution 2.5 Generic license. **B** Flow cytometry

contour plots show surface CD4 and CD8α proteins in pLN. Box plots compile cytometry-based **C** CD8+ T cell counts and **D** CD4/CD8 ratios from pLN and spleens. **E** Flow cytometry contour plots show surface CD4 and CD8α on lymphocytes from bone marrow. Box plots compile CD8+ T cell counts and CD4/CD8 ratios. All ex vivo studies included at least 4 biological replicates per group assayed over at least 3 experiments. Replicate counts and statistical tests for all experiments detailed in Supplementary Data 6.

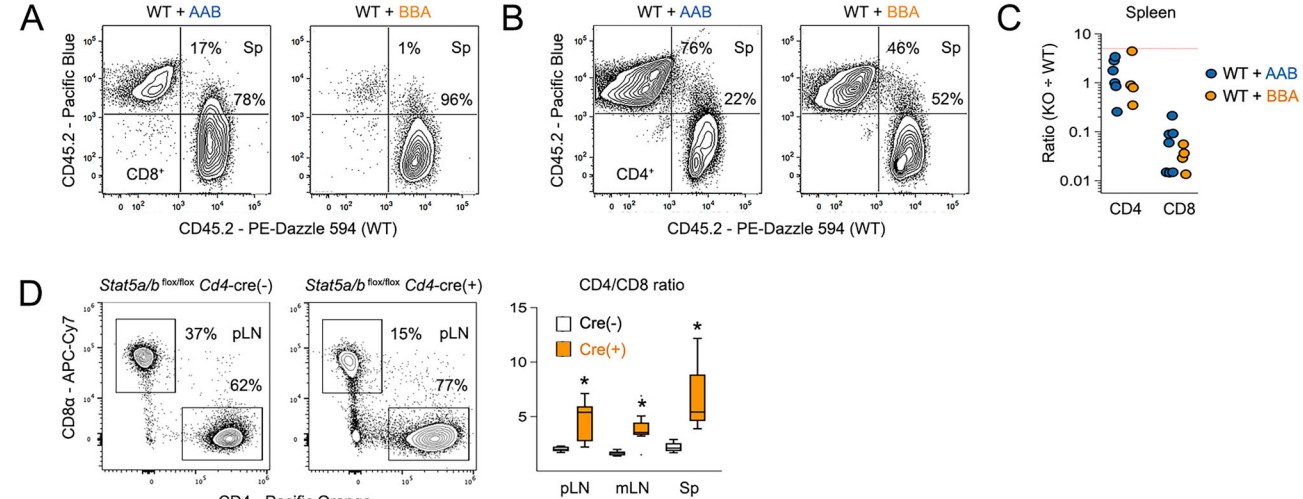

**Fig. 2 | T cell intrinsic requirement for STAT5.** Flow cytometry contour plots show representative **A** CD8[+] and **B** CD4[+] T cell engraftment in spleens of mixed bone-marrow chimeras. WT donor cells are CD45.1 [pos] CD45.2 [neg]; KO donor cells are CD45.1 [neg] CD45.2 [pos]. **C** Scatter plot compiles cytometry-based engraftment data. Each point represents a discrete donor/host pair. Y axis indicates the observed KO to WT ratio in spleens. Red line denotes 5:1 KO to WT starting ratio. **D** Flow cytometry contour plots show surface CD4 and CD8α proteins in pLN of *Stat5* [flox/flox] *Cd4*-Cre[+/−] mice and *Stat5* [flox/flox] *Cd4*-Cre[−/−] littermate controls. Box plot compiles CD4/CD8 ratios across tissues. All ex vivo studies included at least 4 biological replicates per group assayed over at least 3 experiments. Replicate counts and statistical tests for all experiments detailed in Supplementary Data 6.

(Fig. S3A, D). Phosphorylation of tyrosine 694/699, the main instigating event for STAT5 signaling, was also more impacted by STAT5B-deficiency, whether downstream of IL-7 or IL-15 (Figs. 3G and S3B). Given these findings, it is tempting to infer that greater relative abundance explains why STAT5B is dominant over STAT5A in CD8[+] T cells. However, the dramatic phenotypes seen in STAT5B-deficient mice are difficult to reconcile with only 20-40% reductions in total STAT5 levels. Moreover, relative abundance does not explain why CD8[+] T cells are more sensitive to STAT5B-deficiency than CD4[+] T cells, given that total STAT5 content is similar across these lineages (Fig. S3B–E). Ultimately, we conclude that the disparity between T cell phenotypes in STAT5A- and STAT5B-deficient mice likely reflects a combination of asymmetric expression and asymmetric functions, with STAT5B emerging as dominant on both counts.

To further assess the impact of STAT5A and STAT5B deficiencies, we compared transcriptomes directly ex vivo or upon stimulation with IL-7 or IL-15 (Figs. 4A and S4A). These cytokines were chosen because they are prominent in CD8[+] T cell biology and known to activate STAT5 in both naïve and memory states[59]. Using standard surface markers, we first sorted naïve (CD44[low] CD62L[high]), effector (CD44[high] CD62L[low]), and memory (CD44[high] CD62L[high]) CD8[+] T cells from pooled lymph nodes and spleens of WT, AAB, and BBA mice, then cultured them overnight in the presence of IL-7 or IL-15 and made short-read RNA-seq libraries. Next, we called Differentially Expressed Genes (DEG) relative to WT controls using a threshold of >1.5-fold change instead of the more popular >2-fold change to capture as many DEG as possible when genotype-driven effects were subtle. We found a striking disparity; STAT5B-deficient cells (BBA) had many more DEGs than STAT5A-deficient cells (AAB), regardless of state or stimulus (Fig. 4B and Supplementary Data 1, 2). Importantly, DEG called only in STAT5B-deficient cells included many emblematic STAT5 targets (e.g., Bcl2, MYC) linked to cellular pathways known to be STAT5-driven, including several relating to T cell memory and metabolism (Figs. 4B, C and S4B). We also noted that IL-15 mobilized more DEG than IL-7 regardless of genotype or differentiation state, in line with a recent survey of cγ cytokines in immune cells[60], and that it mobilized more DEG in memory than naïve CD8[+] T cells (Fig. 4B). Thus, STAT5B does the heavy lifting downstream of both IL-7 and IL-15, and in both naïve and memory CD8[+] T cells yet resulting STAT5B-driven transcriptional responses vary based on instigating cytokine and/or differentiation state of the responding cells.

Next, we compared DEGs called across cell states and stimuli to determine if any are solely dependent on STAT5A or STAT5B. This analysis revealed that almost every DEG called in AAB cells was also called in BBA cells, indicating that few (if any) are strictly STAT5A dependent (Fig. 4D). By contrast, the vast majority of DEG called in BBA cells were not called in AAB cells (Fig. 4D). We also noted that the degree of induction (i.e., fold change values) differed between "overlapping" and "unique" DEG, the former appearing more labile (Fig. 4E). We interpret that "high mobility" target genes employ both STAT5A and STAT5B, while "low mobility" target genes mainly employ STAT5B, in part, because STAT5A is less available. Immunologically relevant examples of low mobility STAT5 target genes include *Bax*, *Bhlhe40*, and *Nfil3* (i.e., DEG with average fold change <1.3 across STAT5B datasets; Supplementary Data 3).

## STAT5 paralog redundancy in CD8[+] T cells

Upon activation, STAT5A and STAT5B distribute throughout the genome via DNA binding motifs known as "Gamma Interferon Activated Sequences" (GAS motifs)[3]. To determine if they co-localize, we compared genome-wide distributions using ChIP-seq datasets generated in effector CD8[+] T cells stimulated with IL-2[7]. Crucially, these datasets were chosen because absolute numbers of STAT5A- or STAT5B-bound regions, colloquially termed "peaks", are similar (Fig. 5A). We began by identifying peaks enriched over background, then merged those within 100 base pairs of one another to minimize spurious "satellites" that can flank high amplitude peaks (Fig. 5A and Supplementary Data 4). Next, we cross-referenced these merged peak sets to identify genomic regions bound by STAT5A and/or STAT5B. Based on prior studies[7,33], we expected that most captured regions would be bound by both and, indeed, that is what we found. Approximately 2/3 of STAT5A peaks overlapped with STAT5B peaks (and vice versa), while the remaining 1/3 appeared unique to one or the other (Fig. 5A). We then compared summit values and learned that regions bound by both STAT5A and STAT5B had greater means and contained all the highest amplitude peak, suggesting these are robust, high mobility target sites (Fig. 5B). DNA motif analysis also showed that peaks bound by both are far more likely to bear STAT binding motifs than those bound by one or the other (Fig. 5C). Thus, STAT5A and STAT5B tend to co-localize, especially at high traffic sites likely to have functional consequence.

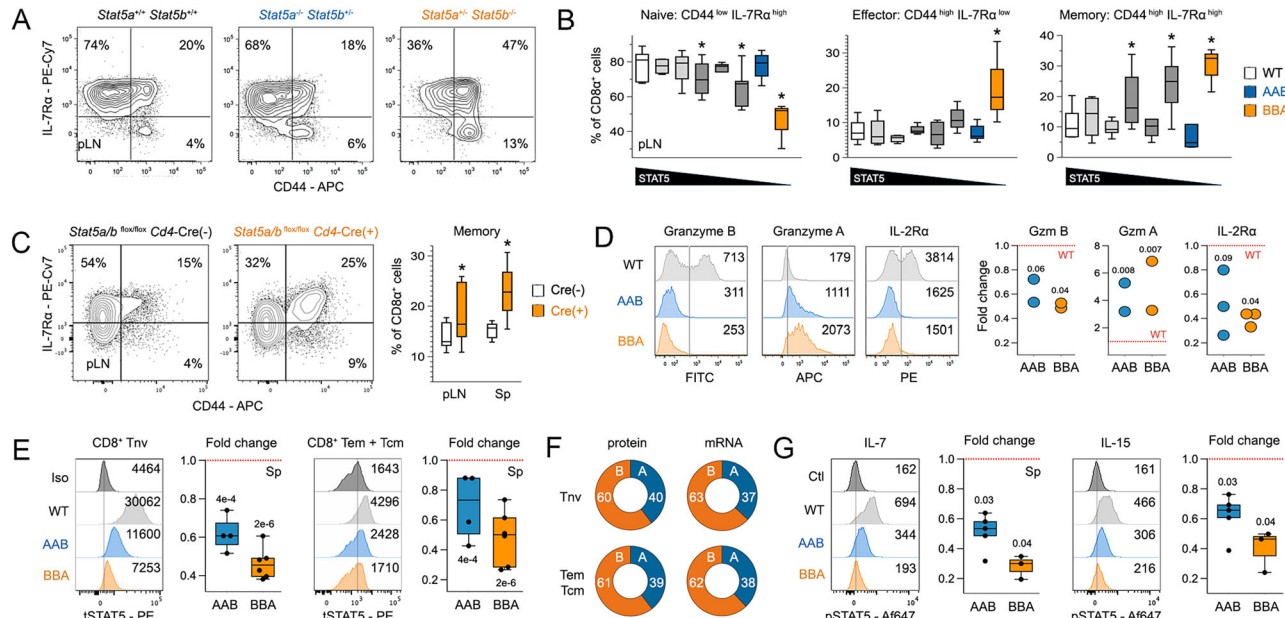

**Fig. 3 | STAT5 deficiency unleashes effector and memory CD8⁺ T cells. A** Flow cytometry contour plots show surface CD44 and IL-7Rα (CD127) on CD8⁺ T cells from pLN. Tnv are defined as CD44 low IL-7Rα high (upper left), Tcm as CD44 high IL-7Rα high (upper right), and Tem as CD44 high IL-7Rα low (lower right). **B** Box plots compile frequencies of naïve, memory, and effector cells in pLN. **C** Flow cytometry contour plots show CD44 and IL-7Rα on CD8⁺ T cells in pLN of *Stat5* flox/flox *Cd4-* Cre⁺/⁻ mice and littermate controls. Box plots compile frequencies of memory cells in pLN and spleens. **D** Flow cytometry histograms show surface or intracellular levels of the indicated proteins in splenic CD8⁺ T cells after 48-h ex vivo culture. Mean fluorescence intensity (MFI) is shown. Box plots compile fold change of MFI values relative to WT controls across biological replicates. **E** Flow cytometry histograms show total STAT5 protein levels in splenic CD44 low naïve (left) and CD44 high effector/memory (right) CD8⁺ cells of the indicated genotypes. Box plots compile

fold change of MFI values relative to WT controls across biological replicates. **F** Donut plots relay the percentage of total STAT5 protein or mRNA accounted for by STAT5A or STAT5B in CD8⁺ Tnv (top row) or Tem/cm cells (bottom row). **G** CD8⁺ Tcm cells from pLN were pulsed with cytokines (or not) for 1 h. Flow cytometry histograms show p-Tyr STAT5 protein levels downstream of IL-7 or IL-15. Box plots compile fold change of MFI values relative to WT controls across biological replicates. **D–G** Red line denotes WT levels (WT = 1). In vitro culture conditions are detailed in Supplementary Data 5. All ex vivo studies included at least 4 biological replicates per group assayed over at least 3 experiments. All in vitro studies included at least 2 biological replicates per group assayed over at least 2 experiments. Replicate counts and statistical tests for all experiments detailed in Supplementary Data 6.

To infer transcriptional effects, we assigned peaks to genes based on genomic proximity. As expected, just over half could be unambiguously assigned, with the majority mapping to a single gene (Fig. 5D). Next, we compared Peak Associated Genes (PAG) and found that, like peaks, a majority of PAG were associated with both STAT5A and STAT5B (Fig. 5D). This was plainly evident at emblematic STAT5 target genes like *Il2ra*, *Bcl2* and *Cish*, where STAT5A and STAT5B peaks closely mirrored (Fig. 5E). We also noted a substantial number of PAG associated only STAT5A or STAT5B but, upon inspection, determined that most of these were not strictly dichotomous. Typically, when a single paralog was assigned to a gene, associated peaks were only slightly under the chosen *p* value threshold, while peaks for the other paralog were only slightly over (Fig. S5). Furthermore, PAG associated with either STAT5A or STAT5B tended to be lower amplitude and contained less STAT motif enrichment, again suggesting that they are not robust targets (Fig. 5B-C).

To further explore redundancy between STAT5 paralogs, we restored STAT5A in STAT5A-deficient CD8⁺ T cells, then measured transcriptomes by RNA-seq, reasoning that if bona fide STAT5A-specific genes exist, then constitutively active STAT5A (CA-STAT5A) should selectively mobilize these and not putative STAT5B-specific genes. For these experiments, naïve CD8⁺ T cells were sorted from AAB mice, stimulated with agonist anti-TCR and anti-CD28 antibodies, then transduced with a retroviral vector expressing constitutively active STAT5A (CA-STAT5A; Fig. 6A). Hundreds of DEG were called relative to cells transduced with "empty" control vector, including many well-known STAT5 targets like BCL2 and GZMB (Fig. 6A). Next, we cross-referenced with ChIP-seq data to segregate DEG based on whether they are proximally bound by STAT5A and STAT5B (5A + 5B), STAT5A alone (5A only), STAT5B alone (5B only) or neither (No STAT5),

per Fig. 5D. Of note, the chosen ChIP-seq dataset was generated in CD8⁺ T cells cultured in vitro with IL-2[7], similar to our retroviral system. Surprisingly, we found that most DEGs were not engaged by STAT5A or STAT5B (Fig. 6B). We interpret this as evidence for indirect regulation (e.g., induction of secondary transcription factors, "piggybacking" on other transcription factors), promiscuous binding due to supra-physiological expression[61] and/or atypical behavior of the constitutively active STAT5A construct used for these studies. Notwithstanding, direct effects were also plainly evident; 182 DEG were bound by both STAT5A and STAT5B, while only 39 were bound solely by STAT5A (Fig. 6B). It is tempting to classify the latter as STAT5A-specific, but we hesitate to do so because a similar number of DEG were engaged by only STAT5B. Also, DEG bound solely by STAT5A or STAT5B had lower fold-change values than those bound by neither or both (Fig. 6C, D). Thus, it may be that some loci are better or even exclusively engaged by STAT5A or STAT5B, but the bulk of evidence suggests that few (if any) bona fide STAT5 targets are solely and strictly dependent on one or the other.

### Evidence for cytokine, lineage, and state-restricted STAT5 activities

Having established that STAT5B is dominant over STAT5A in CD8⁺ T cells, we next compared STAT5B-dependent genes across cytokines and differentiation states. First, we noted a striking distinction between cytokines; IL-15 mobilized nearly 10 times as many DEG as IL-7, regardless of differentiation state (Fig. 7A). This was surprising given IL-7 was slightly better at triggering tyrosine 694/699 phosphorylation in our system and suggests qualitative differences in downstream signaling (Fig. 3G). Accordingly, we found that the character of transcriptional responses also differed; only 35%

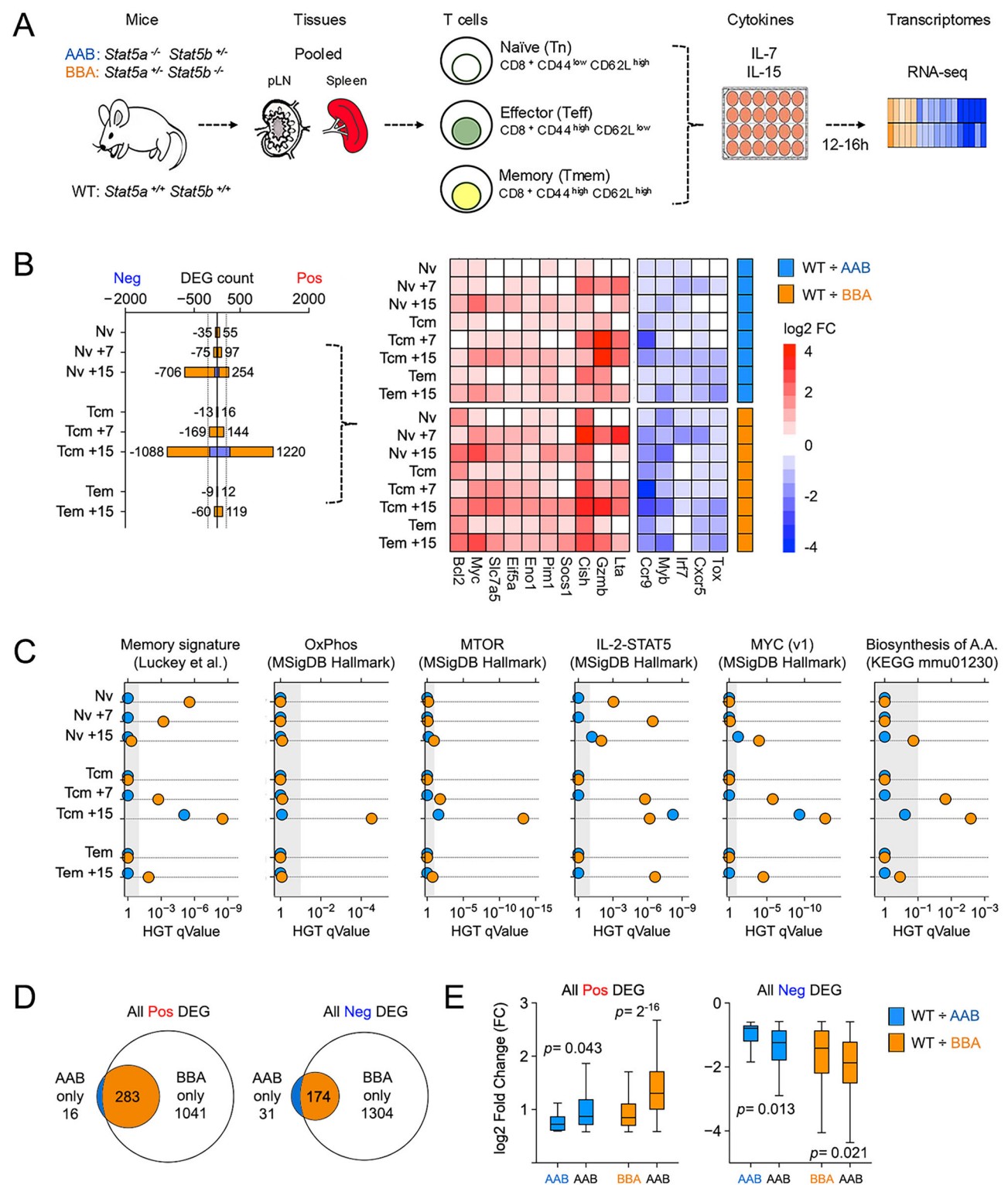

**Fig. 4 | STAT5 paralog asymmetry in CD8⁺ T cells. A** Cartoon outlines an experimental system for transcriptome studies. Mouse line drawing originally published by the Public Library of Science and used here in accordance with the Wikimedia Creative Commons Attribution 2.5 Generic license. **B** Stacked bar plot enumerates Differentially Expressed Genes (DEG) in KO T cells relative to WT controls (blue stack = AAB versus WT; orange stack = BBA versus WT). For reference, dotted vertical lines are drawn at −250 and 250. Heat map shows log2 fold change values for representative DEG across genotypes, cytokines, and cell states. **C** Positively regulated DEG from (**B**) were subjected to hypergeometric testing against the indicated databases. Scatter plots show enrichment *q* values for top STAT5-regulated pathways across genotypes, cytokines, and cell states (blue = AAB

versus WT, orange = BBA versus WT). **D** Positively (left) and negatively (right) regulated DEG were compiled across cytokines and cell states. A Venn plot compares AAB and BBA cells. **E** Box plots show log2 fold change values for DEG classes from (**D**). AAB = DEG is called only in AAB cells, BBA = DEG is called only in BBA, and AAB/BBA = DEG is called in both genotypes. Blue = AAB versus WT, orange = BBA versus WT. All DEG sets are listed in Supplementary Data 1 and usage is detailed in Supplementary Data 2. In vitro culture conditions are detailed in Supplementary Data 5. All in vitro studies included at least 2 biological replicates per group assayed over at least 2 experiments. Replicate counts and statistical tests for all experiments detailed in Supplementary Data 6.

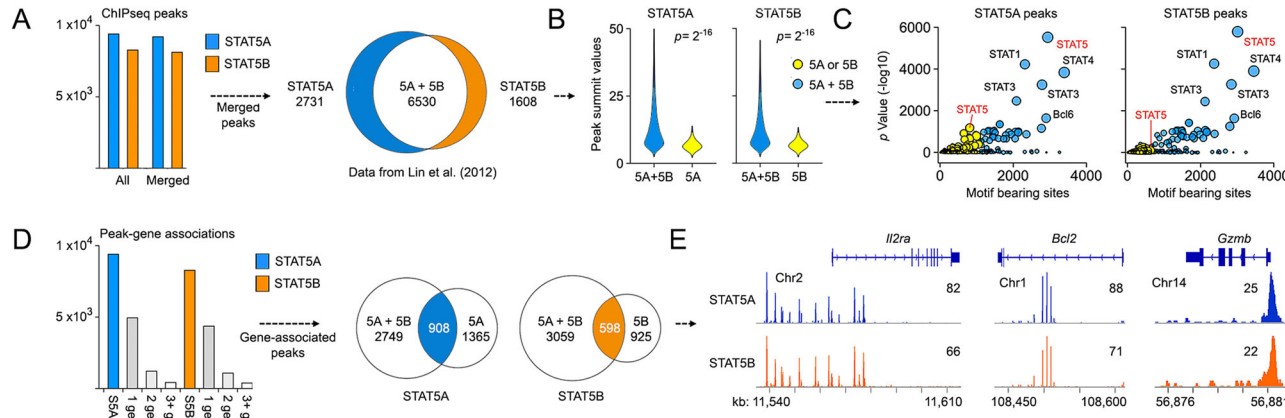

**Fig. 5 | STAT5 paralog redundancy in CD8⁺ T cells. A** Bar plot enumerates DNA regions bound by STAT5A (blue) or STAT5B (orange) pre- and post-peak merge. A Venn plot compares merged regions. Union denotes at least 1 base pair overlap between STAT5A- and STAT5B-bound regions. **B** Violin plots compare peak summit values (i.e., peak amplitude) for STAT5A (left) and STAT5B (right) bound regions across categories defined in (**A**). 5A = regions bound only by STAT5A, 5B = regions bound only by STAT5B, 5A + 5B = regions bound by STAT5A and STAT5B. **C** Scatter plot shows total motif-bearing sites and associated *p* values for STAT5A (left) and/or STAT5B (right) bound regions. Color coding as in (**B**). Point size is proportional to *p* value. **D** STAT5A and STAT5B-bound regions were assigned to genes based on linear proximity. A bar plot enumerates those assigned to one or more genes. A Venn plot compares Peak-Associated Genes (PAG) across categories defined in (**A**). 5A = PAG associated with only STAT5A, 5B = PAG associated with only STAT5B, 5A + 5B = PAG associated with STAT5A and STAT5B. **E** Genome browser histograms show mirroring of STAT5A and STAT5B-bound regions at select loci. All STAT5A and STAT5B-bound regions are catalogued in Supplementary Data 4. Replicate counts and statistical tests for all experiments are detailed in Supplementary Data 6.

of DEG mobilized by IL-7 were also mobilized by IL-15 in naive cells (61 of 172), versus 81% (253 of 311) in memory cells (Fig. 7A and Supplementary Data 1, 2). Moreover, only 17% of DEG mobilized by IL-7 in naive T cells were also mobilized by IL-7 in memory cells (29 of 172), versus 65% for IL-15 (627 of 960; Fig. S6A–C). Thus, cellular state has a major influence on transcriptional activity of STAT5B, with naïve cells exhibiting a more pronounced effect than memory cells. Altogether, we find evidence for: (1) cytokine specificity (e.g., differences between IL-7 and IL-15 in Tnv cells), (2) cell state specificity (e.g., differences between IL-7 in Tnv and Tcm), and (3) stereotypical responses (e.g., similarities between IL-15 and IL-7 in Tcm, and IL-15 in Tnv and Tcm).

To determine if STAT5B activity differs across lineages, we compared transcriptomes from CD4⁺ and CD8⁺ T cells. These studies focused on IL-7 because IL-7Rα is similarly expressed on CD4⁺ and CD8⁺ T cells, unlike IL-15 receptor components IL-2Rβ and IL-15Rα, which are more abundant on CD8⁺ T cells. As before, we performed RNA-seq on naive and memory cells sorted from pooled LN and spleens of AAB and BBA mice, then called DEG relative to WT controls (Fig. 7B and Supplementary Data 1, 2). Next, we cross-referenced with corresponding CD8⁺ T cell datasets and found that: (1) IL-7 mobilizes more DEG in CD4⁺ T cells than in CD8⁺ T cells (Figs. 7B, C), (2) most DEG and pathways mobilized by IL-7 in naive CD8⁺ T cells are also mobilized in naïve CD4⁺ T cells (132 of 172 = 76%; Figs. S6D–F), and (3) relatively few DEGs and pathways mobilized by IL-7 in memory CD8⁺ T cells were also mobilized in memory CD4⁺ T cells (92 of 313 = 29%; Fig. S6D–G). Thus, STAT5 activity does vary across T cell lineages, and, again, the amount of variance depends on cellular state, although, here, memory cells are less affected than naïve.

**A core STAT5 signature for CD8⁺ T cells**
Next, we sought to define a core STAT5 signature evident across cytokines and differentiation states. To begin, we compared DEG called in both naive and memory CD8⁺ T cells downstream of IL-7 or IL-15 and found that they are strikingly distinct; only 15% overlapped (9 of 61; Fig. 7C and Supplementary Data 1, 2). That dichotomy was also evident at the pathway level; naive cells were highly enriched for JAK-STAT signaling, while memory cells were highly enriched for metabolic pathways (Fig. 7A). Next, we devised a composite, or "core", signature by merging pan-cytokine DEG from naive and memory cells. Specifically, we merged all 38 positively

regulated DEG shared between IL-7 and IL-15 in naive CD8⁺ T cells with the top 50 positively regulated DEG shared between IL-7 and IL-15 in memory CD8⁺ T cells, to generate an 85-element signature that accounts for all included variables (Figs. 7C, D and S6H; Supplementary Data 1, 2). Predictably, pathway analysis reflected the composite nature of this core signature; both JAK-STAT signaling and metabolic pathways were enriched (Fig. 7C).

The decision to merge using OR logic rather than distill using AND logic was driven solely by the paucity of DEG shared across cytokines and cell states (9 total; Fig. 7C). The decision to focus on positively regulated DEG was driven by their involvement in emblematic STAT5-regulated pathways, such as JAK-STAT signaling and carbon metabolism[3,6]. However, we also acknowledge that the ability to suppress gene expression is an important, albeit less understood, feature of STAT5 biology and included negative DEGs in the signature tests detailed below (Fig. S7). The decision to focus on the top 50 DEG was driven mainly by the fact that the unabridged set of 130 positive DEG from memory cells far outnumbers the unabridged set of 38 positive DEG from naïve cells and, thus, would have an outsized impact on an OR logic signature. Also, focusing on the top 50 slice based on fold change relative to WT controls ensures that strong DEGs are favored over those barely meeting the chosen DEG call threshold. The goal was to devise a robust core signature that reflects STAT5 activity downstream of both IL-7 and IL-15 in both naïve and memory CD8⁺ T cells.

Several elements of our core signature are known to be induced by STAT5 and/or upstream cytokines in CD4⁺ T cells, including *Il2ra*, *Myc*, and *Slc7a5* (Figs. 7D and S6H)[6]. Thus, to determine if it can distinguish between CD4⁺ and CD8⁺ T cells, we cross-referenced with DEG called in naïve and memory CD4⁺ T cells (from Fig. 7B). Results were clear: most core signature elements were contained within the CD4⁺ T cell gene sets (51/85 = 60% for naïve cells, 67/85 = 79% for memory cells; Fig. 7E). Therefore, despite the fact that it was built exclusively from CD8⁺ T cell data, our core STAT5 signature likely cannot distinguish between STAT5 responses in CD4⁺ and CD8⁺ T cells.

To test whether our core signature can be used as a bioinformatic probe for STAT5 activity, we mined a single-cell RNA-seq (scRNA-seq) dataset composed of CD8⁺ T cells from mice challenged with acute and chronic Lymphocytic Choriomeningitis Virus (LCMV)[62]. After recreating the published UMAP (Figs. 7F and S7), we performed "module score" analysis

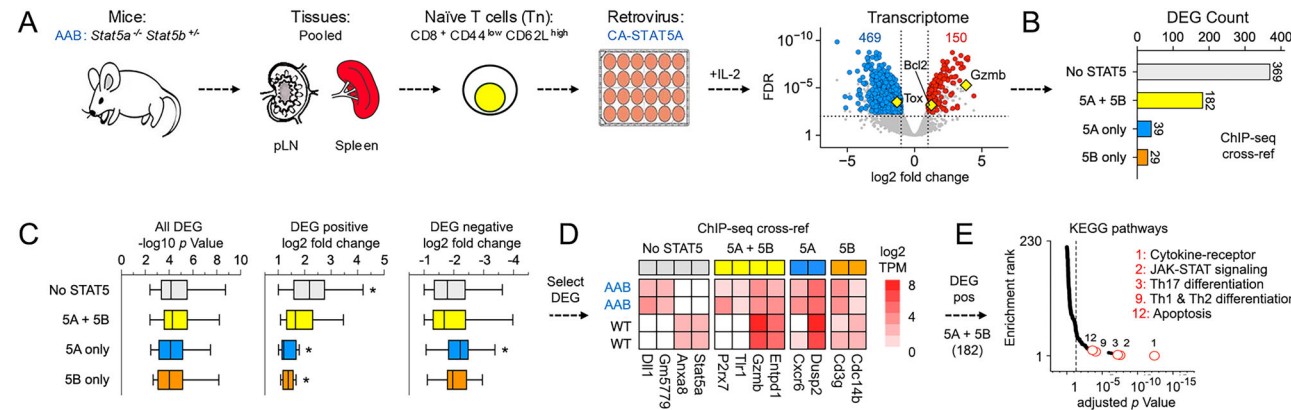

**Fig. 6 | STAT5 paralog specificity in CD8+ T cells. A** *Stat5a−/− Stat5b+/−* CD8+ T cells were transduced with CA-STAT5A, then assayed by RNA-seq. Cartoon outlines experimental system. Volcano plot shows positively (red) and negatively (blue) regulated DEG from pairwise comparison of CA-STAT5 and "empty" control (both in AAB CD8+ T cells). Mouse line drawing originally published by the Public Library of Science and used here in accordance with the Wikimedia Creative Commons Attribution 2.5 Generic license. **B** DEG were cross-referenced with STAT5A or STAT5B Peak-Associated Genes (PAG from Fig. 5D). Bar plot enumerates DEG associated with STAT5A only (5A), STAT5B only (5B), both (5A + 5B), or neither (no STAT5). **C** Box plots compile *p* values and log2 fold change values for DEG segregated as in (**B**). **D** Heat map shows log2 fold change values for representative DEG from each category in WT or AAB cells (each vs. "empty" control). **E** DEG associated with both STAT5A and STAT5B (yellow bar; 182 total) were subjected to hypergeometric testing against the KEGG database. The RankLine plot shows enrichment *p* values and ranks for all captured pathways. Select pathways are noted (rank shown). All DEG sets are catalogued in Supplementary Data 1 and usage is detailed in Supplementary Data 2. In vitro culture conditions are detailed in Supplementary Data 5. All in vitro studies included at least 2 biological replicates per group assayed over at least 2 experiments. Replicate counts and statistical tests for all experiments detailed in Supplementary Data 6.

to determine which regions (if any) are enriched for our core STAT5 signature and/or constituent gene sets (Supplementary Data 1-2). Regarding the latter, we found that all positive gene sets, whether derived from IL-7 or IL-15, naïve or memory CD8+ T cells, were enriched in a single region of the UMAP containing "early effector or exhausted" cells (cluster 6; Figs. 7F and S7B). We interpret that this population had recently encountered antigen and, thus, were experiencing acute STAT5 activity downstream autocrine or paracrine IL-2 responses. By contrast, negative gene sets from naïve CD8+ T cells were enriched in different regions depending on whether driven by IL-7 or IL-15, while those derived from memory CD8+ T cells were not enriched at all (note scales; Fig. 7C). In most cases, narrowing to the top 50 elements based on fold change values enhanced module score performance, reflected in improved enrichment scores and consolidation of UMAP enrichment regions (Fig. S7B, C). Pan-cytokine gene sets from memory CD8+ T cells were far more enriched than those from naïve CD8+ T cells (Fig. S7D) and, crucially, our final core 85-element signature was highly and sharply enriched (Figs. 7F and S7A, D). Thus, our core signature and its constituents may be useful for detecting strong, synchronized STAT5 signaling associated with effector responses, but perhaps not weaker, asynchronous signaling associated with homeostatic responses.

## Discussion

The transcription factor STAT5 is fundamental to lymphocyte biology. However, the relationship between its two paralogs, STAT5A and STAT5B, and the extent to which they are functionally distinct, remains unclear. Here, we engage this longstanding question using a combination of genetic and genomic tools focused on CD8+ cytotoxic T cells, central players in both protective and pathogenic inflammation. Previously, we used a similar approach to demonstrate that STAT5B is dominant over STAT5A in CD4+ T cells and ILCs, where it influences numerous developmental and effector pathways[30,35]. Here, we establish that the "STAT5B > STAT5A" rule holds true across the lymphoid compartment and present evidence for two underlying mechanisms. The first, asymmetric expression, is strongly supported by both protein and transcript data, which shows that STAT5B is twice as abundant in CD8+ T cells as it is in CD4+ T cells and ILCs. The second, non-redundant functions, are supported by multiple data streams, most notably the vast excess of DEG called in STAT5B-deficient T cells relative to STAT5A-deficient T cells, and thousands of genomic regions

associated with STAT5B but not STAT5A. Taken together, these findings resolve the issue of whether STAT5A and STAT5B are redundant or functionally distinct, ultimately leading us to declare that both are true. Stated otherwise, STAT5A and STAT5B are redundant in that they engage many of the same genes and pathways, but STAT5B also has unique, non-redundant functions that are particularly relevant for CD8+ T cells. It is also evident that, while the overarching trends are similar (STAT5B > STAT5A), CD8+ T cells are more sensitive to STAT5B-deficiency than CD4+ T cells. In fact, only CD8+ T cells are depleted in STAT5B-deficient mice, resulting in sharply skewed CD4/CD8 ratios. This finding is in line with prior work on CD8+ T cells[33] and ILCs, which are closely related and also conspicuously depleted in STAT5B-deficient mice[35,63].

STAT5B-deficiency dramatically illustrates the importance of STAT5 signaling in CD8+ T cells. In fact, aberrant CD8+ T cell differentiation - loss of naive quiescence coupled to a shift towards effector and memory states - is a cardinal feature of STAT5B-deficient mice. Given the lack of acute stimulus (e.g., infection or immunization), we interpret that this reflects self-reactivity unleashed by defective Treg-mediated suppression, in line with the idea that the Treg are impaired in STAT5B-deficient mice[30,33]. However, it is important to recognize that STATB is globally deleted in these animals, so we must also acknowledge that CD8+ T cell extrinsic effects likely contribute to the observed phenotypes. Moreover, while prone to effector/memory differentiation, STAT5B-deficient CD8+ T cells cannot elaborate key elements of the cytotoxic effector program (e.g., granzyme expression). Thus, excessive killing likely does not contribute to the attendant kidney disease seen in these mice[30]. Nevertheless, it is clear that STAT5B is critical for cytokine-driven gene expression in CD8+ T cells and that many important cellular pathways are impacted by STAT5B-deficiency at the transcriptional level, including MTOR, MYC, and several involved with metabolism, befitting the key role of STAT5 in that arena[1,6]. Here, again, it is worth noting that GOF STAT5B mutations are more often associated with T cell cancers than STAT5A mutations and appear more potent in driving both cell intrinsic phenotypes and organismal level pathologies[38–40].

Consistent with prior work[33], we demonstrate that STAT5B-deficiency has a far greater impact on gene expression in CD8+ T cells than STAT5A-deficiency. However, genome-wide analysis did not reveal widespread differences in the distribution of STAT5A and STAT5B, in line with the original report of this dataset[7] and recent ChIP-seq studies in CD8+ T cells[33].

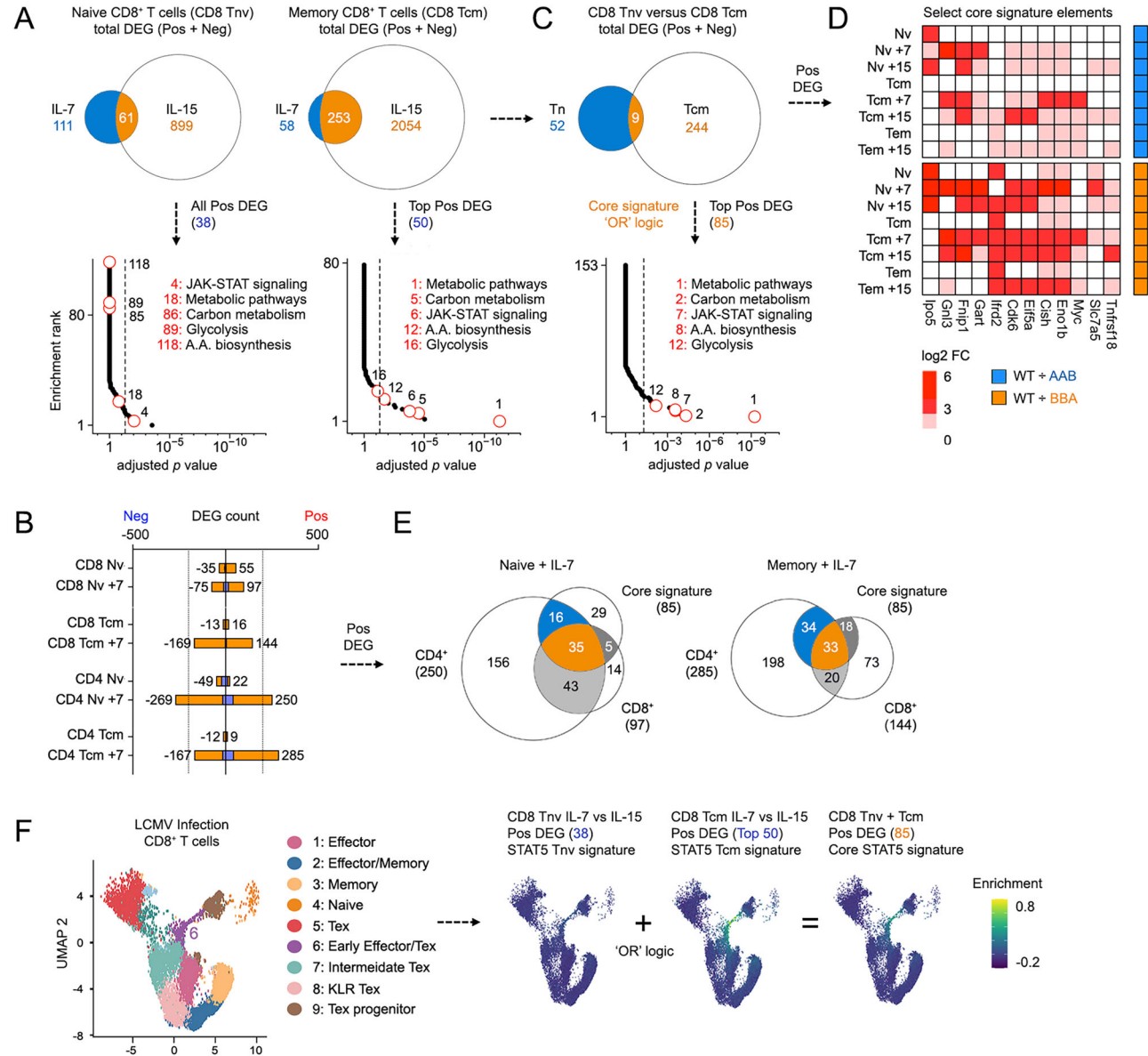

**Fig. 7 | A core STAT5 signature for CD8⁺ T cells. A** Venn plots compare DEG mobilized in naïve (left) or memory (right) CD8⁺ T cells (DEG from Fig. 4). Union represents genes mobilized by both IL-7 and IL-15 in each cell type. Positively regulated "union" DEG from naïve (38 total) and memory (top 50) CD8⁺ T cells were then subjected to hypergeometric testing against the KEGG database. RankLine plots show enrichment $p$ values and ranks for all captured pathways. Select pathways are noted (rank shown). **B** Stacked bar plot enumerates DEG in CD4⁺ and CD8⁺ T cells relative to WT controls (blue stack = AAB versus WT; orange stack = BBA versus WT; CD8⁺ T cell data same as Fig. 4). **C** Venn plot compares "union" DEG from (**A**). RankLine plot shows HGT results for core STAT5 signature genes against the KEGG database. Select pathways are noted (enrichment rank shown). Core STAT5 signature combines all 38 positively regulated genes contained within the IL-7 union set and the top 50 positively regulated genes DEG from the IL-15 union set

(ranked based on fold change). 3 genes are shared, so this yields an 85-element signature. **D** Heat map shows select genes contained in the core STAT5 gene signature (full roster shown in Fig. S6H). **E** Left Venn plot compares the core STAT5 signature to DEG positively regulated by IL-7 in naïve CD4⁺ or CD8⁺ T cells. Right Venn plot compares to memory CD4⁺ or CD8⁺ T cells. **F** scRNA-seq UMAP projection segregates CD8⁺ T cells responding to acute or chronic LCMV infection. Clustering and annotation are as published. Feature plots show module score enrichment for the indicated gene signatures. All gene signatures are catalogued in Supplementary Data 1. In vitro culture conditions are detailed in Supplementary Data 5. All in vitro studies included at least 2 biological replicates per group assayed over at least 2 experiments. Replicate counts and statistical tests for all experiments detailed in Supplementary Data 6.

Instead, we and others have found that STAT5A and STAT5B mostly co-localize and, thus, have similar target ranges. This mirroring is evident across the genome and most obvious at high-amplitude sites decorating bona fide STAT5 targets, like *Il2ra* and *Gzmb*. Furthermore, using a retrogenic system, we established that transcripts mobilized by STAT5A mainly come from loci engaged by both STAT5A and STAT5B, rather than STAT5A alone. Thus, broadly speaking, STAT5A and STAT5B appear to engage the same target genes, making them redundant at the

molecular level. Still, we cannot fully discount the possibility that some genes are subject to only STAT5A or STAT5B. Indeed, we identified: (1) 47 transcripts whose expression was disturbed only in STAT5A-deficient cells, (2) 4339 genomic regions engaged only by STAT5A (25% of all peaks), and (3) 39 genes mobilized by STAT5A and engaged only by STAT5A. However, we also noted that: (1) DEG called only in STAT5A- or STAT5B-deficient cells are "weaker" than those called only in both (i.e., had lower fold change values), (2) ChIP-seq peaks for STAT5A- or STAT5B-specific

regions are "weaker" than those for shared regions (i.e., lower peak amplitudes, higher $p$ values), and (3) a similar number of genes mobilized by STAT5A were engaged only by STAT5B. Therefore, the bulk of evidence suggests that few (if any) genes are controlled solely by STAT5A or STAT5B and that if such genes exist, transcriptional output is likely less than that of genes subject to both.

Given that STAT5B is the dominant paralog, we used our STAT5B datasets to compare STAT5 activities across cytokines, T cell lineages, and differentiation states. Crucially, we learned that despite much commonality, STAT5 responses downstream of IL-7 and IL-15 also vary, with Tnv cells registering greater disparity than Tcm cells. Differences in overall phosphorylation fail to explain cytokine-specific effects, given that IL-15 mobilized more transcripts than IL-7, despite activating less STAT5 on a per cell basis. Alternative explanations include (1) differential phosphorylation; downstream ratios of phospho-STAT5A and phospho-STAT5B may be distinct, (2) kinetic differences; IL-15 may trigger faster or longer p-STAT5 responses, and (3) engagement of parallel signaling pathways like AKT and mTOR. A similar trend emerged when comparing across lineages; IL-7 mobilizes many of the same transcripts in both CD4+ and CD8+ T cells, but also has divergent effects, and, again, the scale of divergence is influenced by differentiation state. Thus, we present evidence for cytokine-specific STAT5 activities (e.g., differences between IL-7 and IL-15 in CD8+ T cells), lineage-specific STAT5 activities (e.g., differences between IL-7 in CD4+ and CD8+ T cells), and cell-state-specific STAT5 activities (e.g., differences between IL-7 in CD8+ Tnv versus Tcm cells).

We also devised an 85-gene core signature comprised of STAT5-regulated genes evident across cytokines and CD8+ T cell differentiation states. To test it, we probed a single-cell RNA-seq dataset that captures CD8+ T cells responding to LCMV infection, a gold standard model for cytotoxic T cell responses[64]. Remarkably, we found that our core signature was enriched in a single cluster of cells annotated as "early exhausted/effectors". We also learned that this enrichment was driven mainly by signature elements derived from memory and IL-15 gene sets rather than naive and IL-7 genesets. Neither the core STAT5 signature nor any of its constituent genesets were enriched in "naive" or "exhausted" clusters, apart from "intermediate exhausted" cells. This is an intriguing result as it suggests that STAT5 signaling is gradually disabled as cells progress from early (or intermediate) to terminal exhaustion. Notwithstanding, it is apparent that our core signature and its constituents is indeed useful for detecting strong synchronized STAT5 responses associated with early T cell activation, as in early effectors. Taken together, our data again affirm the central role of STAT5 in CD8+ T cells and spotlight acute cytokine signaling as a driving force for effector and memory CD8+ T cell responses.

## Methods
### Experimental animals
STAT5 'allele' mice were generated as described[30,35,42]. Briefly, mice lacking the entire STAT5 locus on one chromosome (Stat5a/b+/−) were crossed with mice lacking one allele of STAT5A (Stat5a+/− Stat5b+/+) or STAT5B (Stat5a+/+ Stat5b+/−) to produce 8 allele combinations (Fig. 1A). We refer to each strain according to the alleles that were deleted, except for wild type controls (WT; Stat5a+/+ Stat5b+/+). Thus, AAB mice lack two alleles of STAT5A and one of STAT5B (Stat5a−/− Stat5b+/−; always colored blue), while BBA mice lack two alleles of STAT5B and one of STAT5A (Stat5a+/− Stat5b−/−; always colored orange). Stat5a/b flox/flox Cd4-Cre+/− mice were generated as described[5]. γc−/− Rag2−/− mice were purchased from Taconic Farms. Based on power and size calculations (https://biostat.app.vumc.org/wiki/Main/PowerSampleSize), we determined that >7 mice per group must be included over at least three trials to adequately support the null hypothesis (i.e., no effect). This would be sufficient to detect an effect size of 3 standard deviations using a non-paired t-test or ANOVA. All experiments were unblinded, and animals were allocated to experimental groups based on genotypes. Randomization was not performed, and no animals were excluded. Whenever possible, experimental cohorts were littermates and/or co-housed. Both male and female mice were used and sex matched within

cohorts. No other confounders were controlled against. We have complied with all relevant ethical regulations for animal use. Animals were housed, handled, and euthanized (C0₂ asphyxiation followed by cervical dislocation) in accordance with NIH guidelines, and all experiments were approved by either the NIAMS or the University of Miami Animal Care and Use Committee.

### Cell purification and culture
Tissues were dissected from 8–16-week-old mice and processed to single cell suspensions. Mesenteric lymph nodes (mLN) and peripheral lymph nodes (pLN; inguinal, brachial, axillary, and superficial cervical lymph nodes) were mechanically dissociated through 70 uM cell strainers. Spleens were first mechanically dissociated, then red blood cells depleted by hypotonic lysis (Gibco/Life Technologies). Bone marrow was flushed from the rear femurs, and red blood cells were depleted by hypotonic lysis. CD4+ or CD8+ T cells were enriched by magnetic separation (Miltenyi kits) or cell sorting (as below). Ex vivo culture conditions for all experiments are detailed in Supplementary Data 5.

### Cytometry
For surface antigens, cells were stained and washed directly ex vivo in phosphate-buffered saline supplemented with 0.5% bovine serum albumin and 0.1% sodium azide. For intracellular antigens, naive cells (CD3+, CD8α+, CD44low, CD62Lhigh) were sorted from pooled LN and spleens, stimulated with plate-bound anti-CD3ε and anti-CD28 (1 μg/ml each) in the presence of human IL-2 (100 U/ml; NIH/NCI BRB Preclinical Repository) for 48 h, then fixed and permeabilized with Cytofix/Cytoperm (BD Biosciences). The following fluorochrome labelled antibodies were used: anti-CD4 Pacific Orange (ThemoFisher, 1:500 dilution, clone: RM4-5, cat: MCD0430), anti-CD8α APC-Cy7 (Biolegend, 1:500 dilution, clone 53-6.7, cat 100714), anti-CD3ε PerCP-Cy5.5 (Biolegend, 1:200 dilution, clone 145-2C11, cat 100328), anti-CD45.1 Pacific Blue (Biolegend, 1:500 dilution, clone A20, cat 110722), anti-CD45.2 PE/Dazzle 594 (Biolegend, 1:500 dilution, clone 104, cat 109845), anti-CD44 APC (Biolegend, 1:500 dilution, clone IM7, cat: 103012), anti-CD62L PE-Cy7 (ThemoFisher, 1:500 dilution, clone MEL-14, cat: 25-0621-82), anti-CD127 PE-Cy7 (Biolegend, 1:200 dilution, clone A7R34, cat 135014), anti-GZMA APC (ThemoFisher, 1:200 dilution, clone GzA-3G8.5, cat: 17-5831-82), anti-CD25 PE (Biolegend, 1:500 dilution, clone PC61, cat: 102007), anti-EOMES eflour450 (ThemoFisher, 1:200 dilution, clone Dan11mag, cat: 48-4875-82), anti-TBET PE-Cy7 (Invitrogen, 1:200 dilution, clone: 4B10, cat: 25-5825-80), anti-CD19 Alexa700 (BD, 1:500 dilution, clone: 1D3, cat: 557958). Dead cells were excluded using Live/Dead Aqua (Invitrogen) or Zombie NIR (Biolegend).

### STAT5 measurements
Total STAT5 protein was measured in splenocytes directly ex vivo. Cells were fixed with 2% formaldehyde, permeabilized with 100% methanol, then stained with a rabbit polyclonal IgG that recognizes both STAT5A and STAT5B (sc-835; 1:500 dilution; Santa Cruz Biotechnology) in conjunction with Phycoerythrin labelled goat anti-rabbit IgG for detection (ac-3739; 1:2000 dilution; Santa Cruz Biotechnology). Normal rabbit IgG was used as a negative control (ac-2027; Santa Cruz Biotechnology). For tyrosine-phosphorylated STAT5, splenocytes were treated ex vivo with mouse IL-7 or IL-15 for 1 h (10 ng/ml each; R&D Systems), then fixed with 2% formaldehyde, permeabilized with 100% methanol, and stained with Alexa Fluor 647-labelled anti-human/mouse pY694 STAT5 (Clone 47; 1:50 dilution; BD Biosciences). Ratios of STAT5A and STAT5B were calculated as before[30]. Briefly, mean fluorescence intensity values for Stat5a−/− Stat5b+/− (AAB) or Stat5a+/− Stat5b−/− (BBA) cells were first divided by mean fluorescence intensity values for WT controls (WT) to produce fold-change values, which were then divided by one another, converted to percentages, and presented as donut plots.

Transcriptome data for mouse naive and effector/memory CD8+ T cells was generated in house (Tnv = CD3+, CD4− CD8+ CD44low CD62Lhigh, Tcm = CD3+, CD4− CD8+ CD44high CD62Lhigh, Tem = CD3+,

CD4$^-$ CD8$^+$ Cd44$^{high}$ CD62L$^{low}$) or sourced from the Immunological Genome Project (Tnv = T_8_Nve_Sp, Tem = T8_Tem_LCMV_d180_Sp, Tcm = T8_Tcm_LCMV_d180_Sp, mouse RNA-seq: http://www.immgen.org). Transcriptome data for human naive and effector/memory T cells were sourced from the Immunological Genome Project (CD8 Tnv = Blood_T.8Nve.CD3 + 8 + RA + 62L +, CD8 Tem = Blood_T.8EffMem.CD3 + 8 + RA-62L-, human RNA-seq: http://www.immgen.org), or the Database of Immune Cell Expression, Expression quantitative trait loci (eQTLs) and Epigenomics (DICE: https://dice-database.org, naive cell data only). Normalized expression values (RPKM, FPKM, or TPM) for *Stat5a/STAT5A* and *Stat5b/STAT5B* were averaged, then divided by one another to generate a paralog ratio, which was converted to a percentage and presented as donut plots.

### Bone marrow chimeras
Lineage positive cells were depleted from WT (CD45.1 or CD90.1 congenic), *Stat5a*$^{-/-}$ *Stat5b*$^{+/-}$ (AAB) or *Stat5a*$^{+/-}$ *Stat5b*$^{-/-}$ (BBA) bone marrow by negative selection using Miltenyi Lineage Cell Depletion Kit, supplemented with biotinylated anti-CD335 and anti-CD25. Cells were then washed and re-suspended in PBS before WT, and KO cells were mixed (1:5 WT to KO ratio) and intravenously injected into sex-matched γc−/− Rag2−/− hosts (2 × 10$^5$ total cells per mouse in 300 μl PBS). Engraftment was measured 8–12 weeks later in the spleens.

### RNA-seq
Naive (CD3$^+$, CD4$^+$, or CD8α$^+$, CD44$^{low}$, CD62L$^{high}$), central memory (CD3$^+$, CD4$^+$, or CD8α$^+$, CD44$^{high}$, CD62L$^{high}$) and effector memory (CD3$^+$, CD8α$^+$, CD44$^{high}$, CD62L$^{low}$) T cells were sorted from pooled LN and spleens (>95% purity), then either processed directly ex vivo or cultured for 12–18 h with IL-7 or IL-15 (10 ng/ml each; R&D Systems). Ex vivo culture conditions are further detailed in Supplementary Data 5. 2–4 biological replicates were collected for each experimental group with similar numbers of cells per replicate (20–100 × 10$^3$, depending on group/condition). These were lysed in Trizol reagent and total RNA purified by phenol-chloroform extraction with GlycoBlue as co-precipitant (7–15 μg per sample; Life Technologies). Poly(A)+ mRNA was then enriched by oligo-dT-based magnetic separation, and single-end read libraries were prepared with NEBNext Ultra RNA Library Prep Kit (New England Biolabs). Sequencing was performed with HiSeq 2500 (Illumina), then 50 bp reads (20–50 × 10$^6$ per sample) aligned to the mouse genome build mm10 with *tophat2* (version 2.1.1), assembled with *cufflinks*, and gene-level counts compiled with *htseq-count* (version 2.0.3). To minimize normalization artifacts, genes failing to reach an empirically defined count threshold were purged using *htsfilter* (version 1.32.0). 12–14 × 10$^3$ genes were typically recovered post filtering, regardless of genotype or experimental group. Counts were normalized, and Differentially Expressed Genes (DEG) called by quasi-likelihood F testing using *edgeR* (version 3.34.0). DEG call denotes >1.5 fold pairwise change, and Benjamini-Hochberg (BH) adjusted $p < 0.05$. Transcripts per million (TPM) were compiled with *edgeR*. *clusterprofiler* (version 4.0.0) was used for hypergeometric testing (HGT) against the KEGG, GO, Reactome, Molecular Signatures (MSigDB) databases, or custom genesets (catalogued in Supplementary Data 1; usage detailed in Supplementary Data 2). Heatmaps were rendered with *pheatmap* (version 1.0.13). Venn and Euler plots rendered with *eulerR* (version 7.0.2). All other plots with *ggplot2* (version 3.3.5) or *Datagraph* (version 5.0).

### ChIP-seq
STAT5A and STAT5B ChIP-seq datasets were downloaded from the NCBI sequence read archive via GSE36882[7]. Reads were aligned using *bowtie2* (version 2.4.0), then non-redundant reads were mapped to the mouse genome mm9 using *macs2* (version 2.2.9.1) with 'input' controls as reference for peak calling. *Homer* (version 4.11) was used to annotate peaks and test for TF motif enrichments. Gene proximal peaks were defined as occurring within introns, exons, UTRs, or <10 kb of transcriptional start sites. Genome browser files were rendered with *IGV*. One of two biological replicates was used for all analyses.

### Retroviral transduction
Mutant mouse *Stat5a* (CA-STAT5A = S711F + H299R) was cloned into an MoMLV-based plasmid vector (MigR1) immediately before a dual internal ribosome entry sequence (IRES) and GFP cassette[6]. CA-STAT5A plasmid and pCL-Eco "helper" plasmid were then co-transfected into 293T cells (ATCC) using Lipofectamine (Invitrogen) and virus-containing supernatants collected 48 h later. For transductions, CD8$^+$ T cells were stimulated with plate-bound anti-CD3ε and anti-CD28 (10 μg/ml each) for 24 h, exposed to viral supernatant for 1 h (at 2200 rpm, 18 °C), then cultured for a further 48 h in the presence of human IL-2 (100 U/ml). Viable GFP$^+$ cells were then sorted and processed for RNA-seq.

### scRNAseq
scRNA-seq gene expression (GEX) count matrices were downloaded from NCBI via GSE188670[64]. Downstream analysis was performed with *seurat* using default function parameters, unless otherwise noted. Briefly, ribosomal transcripts were purged, then datapoints excluded if they had >10% mitochondrial transcripts, >15,000 total transcripts, >4000 genes per cell, or <500 genes per cells. Next, individual samples were merged into a single dataset, which was normalized and scaled using *scTransform*. Uniform Manifold Approximation and Projection (UMAP) reduction was then re-created using published XY coordinates at an empirically determined clustering resolution of 0.2. Cluster-defining markers were identified using *FindAllMarkers*. Feature and violin plots were rendered with *seurat*. *clusterprofiler* was used for pathway analysis, specifically hypergeometric testing of the top 400 markers enriched within each cluster (i.e., genes enriched in each cluster relative to all others) against custom STAT5 signature genesets (Supplementary Data 5).

### Statistics and reproducibility
Biological and technical replicates for all experiment detailed in Supplementary Data 6. At least two biological replicates are included for all experimental groups. Statistical variances and distributions were measured by paired $t$ test or Kolmogorov–Smirnov test per Supplementary Data 6. Bonferroni correction was used to account for multiple testing in RNA-seq, ChIP-seq, and pathway analysis datasets. When present, error bars denote standard deviation across >2 biological replicates.

### Reporting summary
Further information on research design is available in the Nature Portfolio Reporting Summary linked to this article.

## Data availability
All data needed to evaluate the conclusions presented in this paper are present herein and/or in the Supplementary Materials. For RNA-seq, gene-level transcript count Supplementary Data are deposited to GEO under accession number GSE319829. Due to logistical problems, raw FASTQ data for the RNA-seq studies are no longer available. Thus, we were able to upload only processed data to GSE319829 (i.e., gene level "counts"). Analysis packages and pipelines used to generate the processed data are described in the "Methods" section, and further details are available upon request.

## Code availability
All analysis pipelines and code are available from the corresponding author upon request.

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

## Acknowledgements

We thank members of the O'Shea, Villarino, and Malek labs for discussions, Gustavo Gutierrez-Cruz for sequencing support, and the NIAMS Flow Cytometry Group for cell sorting. Research in this publication was performed with support from the NIAMS intramural research grant 1 ZIA AR041159-09 (JJO); University of Miami, Department of Microbiology and Immunology grant PG013596 (AVV); University of Miami, Sylvester Comprehensive Cancer Center grant PG012707 (AVV). This work uses resources provided by the Flow Cytometry Shared Resource (SCR_022501) and Onco-Genomics Shared Resource (SCR_022502) of the Sylvester Comprehensive Cancer Center at the University of Miami Miller School of Medicine, which is supported by a National Cancer Institute Cancer Center Support Grant (CCSG) P30-CA240139 and State of Florida Bankhead Coley Research Infrastructure Grant 8BC09. This research was also supported by the Intramural Research Programs of the National Institute of Diabetes and Digestive and Kidney Diseases (NIDDK) and the National Institute of Arthritis and Musculoskeletal and Skin Diseases (NIAMS) within the National Institutes of Health (NIH). The contributions of the NIH author(s) are considered Works of the United States Government. The findings and conclusions presented in this paper are those of the author(s) and do not necessarily reflect the views of the NIH or the U.S. Department of Health and Human Services.

## Author contributions

Conceptualization: A.V.V. Methodology: A.V.V. Investigation: A.V.V., S.R., A.M.F. Data analysis: M.D., C.A., N.I., and L.N. Visualization: A.V.V., M.D., and L.N. Funding acquisition: A.V.V. and J.O.S. Project administration: A.V.V. Supervision: A.V.V., J.O.S., and L.H. Writing—Original draft: A.V.V. Writing—Review & editing: S.R., M.D., L.H., A.V.V., and J.O.S.

## Competing interests

J.J. O'Shea and the NIH hold patents related to therapeutic targeting of Jak kinases and have a Collaborative Research Agreement and Development Award with Pfizer. All other authors declare no competing interests.
