## [Transparent Peer Review File · Communications Biology]

Asymmetry and redundancy of STAT5 paralogs across CD8+ T cell differentiation states

Corresponding Author: Dr Alejandro Villarino

Version 0:

Reviewer comments:

Reviewer #1

(Remarks to the Author)

Summary:

The transcription factor STAT5 is a key regulator of immune cell responses, downstream of cytokines including IL-2, IL-7, and IL-15. It plays important roles in regulating T cell homeostasis, effector responses, and memory cell formation. STAT5 has two paralogs, STAT5A and STAT5B, which have been shown to exert both unique and redundant effects in prior T cell studies, yet the specific roles of these paralogs across T cell types (CD4 vs. CD8) and mature T cell subsets are yet to be determined. Further, how STAT5-signaling cytokines exert differential effects while sharing STAT5 as a downstream transcription factor is incompletely understood.

Here, Ristin et al. leverage STAT5 'allele' mice, which exhibit diminishing STAT5 alleles, to examine paralog-specific and shared roles for STAT5A and STAT5B in regulating CD8 T cell biology. This 'titration' of STAT paralogs offers unique insights into the role of each paralog in regulating CD8 T cell compartments, with an emphasis on naïve and memory CD8 T cells, including a core set of genes regulated by STAT5 across CD8 T cell subsets. Such information is valuable in the context of our understanding of CD8 T cell biology, and may help to set the stage for potential modulation of STAT activity for therapeutic benefit. The manuscript is well-written and well-controlled. However, enthusiasm for the manuscript in its current form is tempered by several limitations, detailed in full below:

Major points:

1. The authors utilize a combination of germline knockouts, mixed bone marrow chimeras, and conditional knockouts to assess various aspects of STAT paralog contributions to CD8 T cell biology. It is this reviewer's understanding that germline knockout cells were utilized for the sequencing studies. However, while the STAT paralog-dependent alterations in CD8 T cell composition were separately confirmed by the authors to be T cell-intrinsic, STAT5-mediated effects on other immune cell populations may affect polarization of assessed populations (particularly memory CD8 T cells) for sequencing. This should be addressed.
2. A limitation of the study is the lack of quantification of STAT5A and STAT5B protein in the generated mice, as it does not allow the authors to rule out the possibility of compensatory elevated expression in heterozygous mice (i.e. upregulation of B in AAB, and A in BBA). While the relative abundance of each protein is inferred based on the drop in overall STAT5 in specific knockouts, staining for specific paralogs would definitively provide this information. At minimum, expression levels for each paralog should be assessed in both naïve and memory CD8 T cell populations from these mice, to ensure that it is indeed consistent with the genotype.
3. The authors infer paralog-specific genes from sequencing data based on both genotype and published ChIP-seq datasets-- highlighting that reliable antibodies for the individual paralogs are available (at least for IP). Because the relative abundance, and not just identity, of STAT5 paralogs may affect enrichment at specific target sites, the authors should strongly consider performing ChIP/ChIP-seq using T cells from the STAT5 'allele' mice to strengthen conclusions drawn regarding specific gene targets for each paralog.
4. It is possible that individual cytokines (or cell types) exhibit a preference for activation and dimerization of a particular paralog. It would clarify the findings to assess not only expression (as noted above), but also the tyrosine phosphorylation (activation) status of each paralog downstream of IL-2, IL-7, and IL-15. This may also provide insight into a potential mechanism by which these cytokines exert their specific effects.

5. RNA-seq findings would be strengthened by including at least 3 biological replicates per condition for analysis.

6. It is unclear to this reviewer whether the “merging” of upregulated DEGs to generate the ‘core’ list was through identification of genes that were upregulated in BOTH naïve and memory CD8 T cells, or whether all upregulated DEGs from the two were simply combined. Further, use of only 50 (or 35) of the most up-regulated genes severely limits the breadth of the core gene analysis. The authors should update their core identification method to consider all significant DEGs, only include DEGs that are shared between groups, and include an analysis of down-regulated DEGs (as STAT5 is also known to function as a transcriptional repressor, this feels like a missed opportunity for some really interesting findings). Given the limited number of DEGs used in the subsequent single-cell analysis, the associated statements in the results and discussion do not currently seem to be supported.

Minor points:

1. The authors utilize IL-2, IL-7, and IL-15 to stimulate CD8 T cell populations at different points throughout the manuscript. It is well-established that these cytokines exert both shared and unique effects, which are T cell type- and differentiation-state specific. Care should be taken to clearly state why each cytokine was used for each study (e.g. why IL-7 and IL-15 were used for RNA-seq studies, yet not IL-2).

2. The authors should consider including information regarding DEGs that were up- or down-regulated >2-fold or >3-fold; 1.5-fold is a relatively modest increase. This information may also help with core gene set identification, as DEG identification would be more stringent.

3. For Figure 3D, an isotype control antibody stain should be included in each histogram plot. Further, repeated analyses should be enumerated and statistical analysis should be performed in order for the stated conclusions to be adequately supported.

Reviewer #2

(Remarks to the Author)

This manuscript by Ristin, Dalzell and colleagues examines the redundant and unique roles of paralogs STAT5A and STAT5B in CD8+ T cells, their differentiation states and responses to cytokines. This study builds on previous work published by the group, which used the same genetic mouse models of STAT5A/B deficiency to explore the roles and dosage effects of the two STAT5 proteins in CD4+ T cells (PMID: 26999798). Here, they reveal that STAT5B is largely dominant in its regulation of CD8+ T cell states, most likely due to its higher abundance rather than possessing a unique functional capacity over STAT5A. They also develop a core STAT5 gene signature that, while unable to distinguish CD8+ T cells from CD4+ T cells due to considerable overlap, could be used to define strong STAT5 activity present selectively in early effector/exhausted CD8+ T cells.

The following minor points should be addressed:

- Figure 1B, the bottom left-most facs plot has an error in the genotype heading, reading ‘Stat5a-/- Stat5b-+/+’ (a minus in the Stat5b genotype that shouldn’t be there).
- Figure 2A,C: the authors state that the WT/BBA chimera has “even more” reduction of splenic CD8+ T cells than the WT/AAB chimera. Whilst the selected plot shown in panel A reflects this strongly, statistics should be performed on the comparison of quantified ratios in panel C, as it does not really appear that there is a difference overall. The authors also state that the 1:5 ratio for the CD4+ T cells in the chimeras is “largely preserved”. While it is certainly closer than the CD8+ T cells, it is also reduced and the wording of the text should better reflect this.
- It would be helpful for the reader if it is indicated in Figure 3B (within the graph) that these data are from pLN, just as it is indicated that data in Figure S2A are from spleen. Also, indicating BM for the data in Figure S2B. It is easier than looking through figure legends.
- The memory subset graph in Figure 3C is duplicated (identical) in Figure S2C. It should be removed from the supplementary, showing only the additional naïve and effector subsets there. Or all could be simply shown together in Figure 3C.
- What do the dotted vertical lines in Figure 4B (left) represent? In the text it is stated that DEG called only in STAT5B deficient cells include targets such as Bcl2 and Myc, yet in the heatmap in Figure 4B (right) these and all other shown targets seem to be similarly dysregulated across both STAT5A and STAT5B deficient conditions. This is unclear.
- Figure 4D: the label “AAB only” is missing from the right ‘neg’ DEG diagram.
- In the text, Figure 5D and 5E are mistakenly referred to as 4D and 4E.
- Figure 7A: the text states that here “STAT5B-driven transcriptomes” are compared across cytokines and differentiation states, but it is difficult to assess now which mouse genotypes are being assessed. One would assume only the BBA vs WT but this is not stated anywhere. The use of orange and blue in panel A and B is also confusing, as here it appears to rather refer to the cytokines or differentiation states, not the usual mouse genotypes.
- It would be worth noting in the discussion that it has been previously shown that oncogenic (constitutively active) forms of STAT5 are much more promiscuous with respect to DNA binding and regulation of gene expression at non-canonical loci than their endogenous ‘normal’ counterparts, which is likely why so many genes were differentially regulated in this system but not found in the integrated ChIP-seq dataset. They may not necessarily be indirect effects. However, indirect effects (gene regulation via protein interaction partners and not via direct STAT5 DNA binding) could also be playing a role in the

normal system – STAT5B may mediate such effects to a greater extent than STAT5A, which cannot be distinguished as STAT5B-only regulated genes by using ChIP-seq data.

- In general, the figures are quite small and are hard to read without getting very close or zooming in. The figures, or at least the font/text, should be made larger.
- There are some minor spelling errors/missing words throughout that should be checked carefully (examples are hard to give without line numbers). In the first paragraph of the introduction, do the authors mean 'systematic' autoimmunity, or rather 'systemic'?

Reviewer #3

(Remarks to the Author)

The manuscript by Ristin, Dalzell et al. investigates the distinct yet overlapping roles of STAT5A and STAT5B in CD8⁺ T cells, using mouse models to dissect their contributions to T cell biology. The authors demonstrate that although STAT5A and STAT5B share molecular homology, they exhibit functional asymmetry, with STAT5B playing a dominant role. This work builds on previous studies that have shown similar findings in other cell types, but extends the analysis in greater depth within the context of CD8⁺ T cells. The data provided show that STAT5B is more abundant, contributes disproportionately to STAT5-dependent processes, and exhibits both cytokine- and cell state-specific functions.

This is a well-conceived study that makes a clear contribution to understanding STAT5 biology in CD8⁺ T cells. The distinction between redundancy and asymmetry, with STAT5B dominance, is conceptually significant and directly relevant to potential therapeutic approaches.

Suggestions:

1. While the manuscript clearly highlights the importance of STAT5 signaling in CD8⁺ T cells, the review of the existing literature could be strengthened. In particular, more attention should be given to recent findings that connect STAT5 biology to cancer development and progression. Integrating these cancer-related insights would position the current study in a broader biological and translational framework. Doing so would also underscore the relevance of asymmetric paralog function not only in T cell biology but also in oncogenic signaling, thereby enhancing the manuscript's impact for both immunology and cancer biology audiences.

2. Finally, given the translational emphasis, a discussion of whether these findings are conserved in human cells would further strengthen the manuscript and broaden its impact.

Version 1:

Reviewer comments:

Reviewer #1

(Remarks to the Author)

The authors have sufficiently addressed my concerns. I now find this manuscript acceptable for publication in Communications Biology.

Reviewer #2

(Remarks to the Author)

All concerns have been adequately addressed.

We thank the reviewers for their thoughtful and thorough critiques of our work. In response, we added substantial new data and heavily revised the text. These additions strongly support our original findings and have improved both the clarity and impact of the manuscript, which we hope is now suitable for publication in *Communications Biology*. Below are point-by-point responses to reviewer comments.

Reviewer 1 point-by-point (major)

1. "...STAT5-mediated effects on other immune cell populations may affect polarization of assessed populations (particularly memory CD8 T cells) for sequencing. This should be addressed."

Reviewer 1 is correct in noting that T cell extrinsic effects may account for at least part of the reported T cell phenotypes, given our STAT5 allele mice are germline KOs. We sought to address this caveat in the original manuscript with: (1) mixed bone marrow chimeras, and (2) conditional KO mice where STAT5 (both A and B) is selectively deleted in T cells. However, as noted by Reviewer 1, the fact that both sets of studies point towards T cell intrinsic effects does not discount cell extrinsic effects in the germline KO mice. We now plainly acknowledge this possibility in both the Results and Discussion sections (quoted below).

Results: "Together, these data affirm that that CD8+ T cells are more impacted by STAT5 deficiency than CD4+ T cells and, crucially, that the CD4/CD8 skewing seen in our STAT5 allele mice likely reflects T cell intrinsic functions. However, it is still true that STAT5 signaling is globally depressed and that several (if not all) of these mouse lines have chronic, baseline inflammation (30), so we must acknowledge that cell-extrinsic 'knock-on' effects likely contribute to the observed CD8+ T cell phenotypes."

Discussion: "However, since STATB is globally deleted in these animals, we must also acknowledge that, CD8+ T cell extrinsic effects likely contribute to the observed phenotypes."

2. "A limitation of the study is the lack of quantification of STAT5A and STAT5B protein in the generated mice.... staining for specific paralogs would definitively provide this information. At minimum, expression levels for each paralog should be assessed in both naïve and memory CD8 T cell populations from these mice, to ensure that it is indeed consistent with the genotype."

The overarching thesis of the original manuscript was that STAT5B is dominant over STAT5A, in large part, because it is more abundant. Thus, as recognized by Reviewer 1, quantification of STAT5A versus STAT5B is key supporting evidence for the proposed mechanism. We have now fortified this finding in three ways. First, we used our original pan-STAT5 assay (i.e. cytometry-based measurement of total STAT5 protein) to calculate the ratio of STAT5A and STAT5B protein in naïve and effector/memory CD8+ T cells (Fig. 3E-G and Fig. S3B-C,E). Second, we used our transcriptome datasets to calculate the ratio of *Stat5a* and *Stat5b* transcripts in naïve and memory CD8+ T cells (Fig. 3F and Fig. S3A-B,D). Third, we used published transcriptome datasets to confirm that it holds true in human CD8 T cells (Fig. S3A,D). Overall, there was good accord between the protein and mRNA data in demonstrating that STAT5B is from 1.5 to 3 times more abundant than STAT5A in CD8+ T cells.

We also now plainly acknowledge that, "it is still difficult to reconcile the dramatic phenotypes seen STAT5B-deficient mice with 20-40% reductions total STAT5", and that, "disparity between T cell phenotypes in STAT5A- and STAT5B-deficient mice likely reflects a combination of asymmetric expression and asymmetric functions, with STAT5B emerging as dominant on both counts"

The revised manuscript clearly declares this balanced interpretation in the Abstract, Results and Discussions (quoted below), with details on paralogs measurements, relative abundance calculations and datasets provided in the Methods.

We also note here that we fully agree with Review 1 in that the most straight forward way to compare STAT5A and STAT5B would be to measure them individually. This is how we did it for the newly added RNA-seq data but not the protein data, which relies on pan-STAT5 measurements. Ultimately, we believe that comparing two different proteins based on two different antibodies is confounded by inevitable variance in antibody performance (i.e. differences in affinity and/or avidity). That is why we have chosen to infer paralog abundance

based on performance of a single pan-STAT5 antibody that recognizes a single homologous epitope, rather than by measuring them individually.

Abstract: “and present evidence that it exhibits paralog-specific effects”

Results: “STAT5B accounts for approximately two-thirds of total STAT5 in CD4+ T cells (30). Thus, we next asked if asymmetric expression is also evident in CD8+ T cells. First, we compared total STAT5 protein levels (STAT5A + STAT5B) in CD44 low naïve and CD44 high effector/memory CD8+ T cells from WT, AAB and BBA mice. As with CD4+ T cells, we found that STAT5B deficiency clearly had greater impact and, in turn, calculated that it accounts for about 60% of the total STAT5 pool in naïve and effector/memory CD8+ T cells (Fig. 3E). Similar disparity was also evident at transcriptional level where, again, Stat5b accounts for 60-70% of total Stat5 mRNA (Fig. 3F) and holds true whether transcriptomes were generated in-house (detailed below) or mined from public databases (Fig. 3F and Fig. S3A-C), and whether sourced from mouse or human T cells (Fig. S3A,D). Phosphorylation of tyrosine 694/699, the main instigating event for STAT5 signaling, was also more impacted by STAT5B-deficiency, whether downstream of IL-7 or IL-15 (Fig. 3G and Fig S3B). Given these findings, it is tempting to speculate that greater relative abundance explains why STAT5B is dominant over STAT5A in CD8+ T cells. However, the dramatic phenotypes seen in STAT5B-deficient mice are difficult to reconcile with only 20-40% reductions in total STAT5 levels. Moreover, relative abundance does not explain why CD8+ T cells are more sensitive to STAT5B-deficiency than CD4+ T cells, given total STAT5 content is similar across these lineages (Fig. S3B-E). Ultimately, we conclude that disparity between T cell phenotypes in STAT5A- and STAT5B-deficient mice likely reflect a combination of asymmetric expression and asymmetric functions, with STAT5B emerging as dominant on both counts.”

Discussion: “Here, we establish that the ‘STAT5B > STAT5A’ rule holds true across the lymphoid compartment and present evidence for two underlying mechanisms. The first, asymmetric expression, is strongly supported by both protein and transcript data which shows that STAT5B is twice as abundant in CD8+ T cells, as it is in CD4+ T cells and ILCs. The second, non-redundant functions, is supported by multiple data streams, most notably the vast excess of DEG called in STAT5B-deficient T cells relative to STAT5A-deficient T cells, and thousands genomic regions associated with STAT5B but not STAT5A. Taken together, these findings resolve the issue of whether STAT5A and STAT5B are redundant or functionally distinct, ultimately leading us to declare that both are true. Stated otherwise, STAT5A and STAT5B are redundant in that they engage many of the same genes and pathways but STAT5B also has unique, non-redundant functions that are particularly relevant for CD8+ T cells.”

Methods: Details on all data sets, and methods for STAT5 paralog measurements and calculations.

3. “ Because the relative abundance, and not just identity, of STAT5 paralogs may affect enrichment at specific target sites, the authors should strongly consider performing ChIP/ChIP-seq using T cells from the STAT5 ‘allele’”

Like Reviewer 1, we expect that genome-wide distribution would be more impacted by STAT5B deficiency than STAT5A deficiency in CD8+ T cells. Indeed, that is what we found when we performed pan-STAT5 ChIP-seq (using the same antibody that we use for cytometry) in CD4+ T cells (Villarino et al., eLife 2014). Far fewer STAT5 peaks were detected in STAT5B-deficient than STAT5A-deficient CD4+ T cells and, interestingly, those that remained mapped to regions with high amplitude binding in WT controls (i.e. robust target sites). Ultimately, we believe that, while high amplitude sites may differ, results would be similar in CD8+ T cells. Thus, we decided to mine published data sets for STAT5A and STAT5B instead, citing all major caveats about cross-referencing separate ChIP-seq assays. Here it is also important to note that the pan-STAT5 antibody that we used previously clearly does not perform as well as the anti-STAT5A and anti-STAT5B antibodies used in the published ChIP-seq studies, given the latter produced >10X more peaks.

4. “ It is possible that individual cytokines (or cell types) exhibit a preference for activation and dimerization of a particular paralog. It would clarify the findings to assess not only expression (as noted above), but also the tyrosine phosphorylation (activation) status of each paralog...”

As noted by Reviewer 1, the balance of phosphorylation between STAT5A and STAT5B may vary downstream of different cytokines, providing an avenue for divergent cytokine functions and further explaining phenotypic differences between STAT5A- and STAT5B-deficient T cells. As far as we know, there are no antibodies that exclusively detect phospho-STAT5A or phospho-STAT5B, and we were unable to find published evidence that phosphorylation of STAT5A and STAT5B has been compared downstream of IL-7 or IL-15, the cytokines in question here. Therefore, we chosen to infer relative contributions of STAT5A and STAT5B using a single antibody that recognizes both phospho-STAT5A and phospho-STAT5B. Like for total STAT5, we found that STAT5B accounts for about 60% of phospho-STAT5 downstream of either IL-7 or IL-15, which we interpret as evidence for balanced phosphorylation given A:B ratios are the same pre- and post-cytokine. We also now report a notable difference in overall STAT5 phosphorylation downstream of IL-7 and IL-15 and better acknowledge that STAT5 signaling downstream of IL-7 and IL-15 may be both qualitatively and quantitatively distinct.

The preceding findings and interpretations on STAT5 phosphorylation are now included in the manuscript, as quoted below.

Results: "...we noted a striking distinction between cytokines; IL-15 mobilized nearly 10 times as many DEG as IL-7, regardless of differentiation state (Fig. 7A). This was surprising given IL-7 was slightly better at triggering tyrosine 694/699 phosphorylation in our system and suggests qualitative differences in downstream signaling (Fig. 3G)."

Discussion: "Differences in overall phosphorylation fail to explain cytokine specific effects given we IL-15 mobilized more transcripts than IL-7 despite activating less STAT5 on a per cell basis. Alternative explanations include, (1) differential phosphorylation; downstream ratios of phospho-STAT5A and phospho-STAT5B may be distinct, (2) kinetic differences; IL-15 may trigger faster or longer p-STAT5 responses, and (3) engagement of parallel signaling pathways like AKT and mTOR."

5. "RNA-seq findings would be strengthened by including at least 3 biological replicates per condition for analysis"

It is irrefutable that more replicates would improve the statistical power of our studies and, in turn, strengthen conclusions. Indeed, it was our intention to have 3-4 biological replicates per group for all RNA-seq studies but, regrettably, sequencing libraries for one replicate set did not pass quality control and, thus, had to be omitted. This left us with 2-3 replicates per group which, due to logical issues, we are not in position to augment. Thus, while we fundamentally agree with Reviewer 1 that statistical power of these studies stands to be improved, we must stand by our original findings and maintain that, while suboptimal, data from groups with only 2 replicates is reliable. It is also worth noting that, in our experience, false negatives are a greater concern than false positives for minimally powered RNA-seq studies; typically, inter-replicate variance is a major hurdle when calling DEG with fewer replicates using the edgeR analysis package, provided consistent direction of effect.

6. "It is unclear to this reviewer whether the "merging" of upregulated DEGs to generate the 'core' list was through identification of genes that were upregulated in BOTH naïve and memory CD8 T cells, or whether all upregulated DEGs from the two were simply combined. Further, use of only 50 (or 35) of the most up-regulated genes severely limits the breadth of the core gene analysis. The authors should update their core identification method to consider all significant DEGs, only include DEGs that are shared between groups, and include an analysis of down-regulated DEGs"

Reviewer 1 makes several insightful points about decisions made in devising our STAT5 gene signatures. Regrettably, we did not adequately explain those decisions, nor the choices behind them, in the original manuscript. To rectify, we completely rewrote the relevant section of the manuscript to include detailed explanation of how all signatures were devised and added complementary analyses for negative DEG (quoted below).

"...Next, we devised a composite, or 'core', signature by merging pan-cytokine DEG from naive and memory cells. Specifically, we merged all 38 positively regulated DEG shared between IL-7 and IL-15 in naive CD8+ T cells with the top 50 positively regulated DEG shared between IL-7 and IL-15 in memory CD8+ T cells, to

generate an 85-element signature that accounts for all included variables (Fig. 7C-D; Fig. S6H, Table S1-S2). Predictably, pathway analysis reflected the composite nature of this core signature; both JAK-STAT signaling and metabolic pathways were enriched (Fig. 7C).

The decision to merge using OR logic rather than distill using AND logic was driven solely by the paucity of DEG shared across cytokines and cell states (9 total; Fig. 7C). The decision to focus on positively regulated DEG was driven by their involvement in emblematic STAT5-regulated pathways, such as JAK-STAT signaling and carbon metabolism (3, 6). However, we also acknowledge that the ability to suppress gene expression is an important, albeit less understood, feature of STAT5 biology and included negative DEGs in the signature tests detailed below (Fig. S7). The decision to focus on the top 50 DEG was driven mainly by the fact that the unabridged set of 130 positive DEG from memory cells far outnumbers the unabridged set of 38 positive DEG from naïve cells and, thus, would have outsized impact on an OR logic signature. Also, focusing on the top 50 slice based on fold change relative to WT controls ensures that strong DEG are favored over those barely meeting the chosen DEG call threshold. The goal was to devise a robust core signature that reflects STAT5 activity downstream of both IL-7 and IL-15 in both naïve and memory CD8+ T cell.

Several elements of our core signature are known to be induced by STAT5 and/or upstream cytokines in CD4+ T cells, including *Il2ra*, *Myc* and *Slc7a5* (Fig. 7D, Fig. S6H)(6). Thus, to determine if it can distinguish between CD4+ and CD8+ T cells, we cross-referenced with DEG called in naïve and memory CD4+ T cells (from Fig. 7B). Results were clear: most core signature elements were contained within the CD4+ T cell gene sets (51/85 = 60% for naïve cells, 67/85 = 79% for memory cells; Fig. 7E). Therefore, despite the fact that it was built exclusively from CD8+ T cell data, our core STAT5 signature likely cannot distinguish between STAT5 responses in CD4+ and CD8+ T cells.

To test whether our core signature can be used as a bioinformatic probe for STAT5 activity, we mined a single-cell RNA-seq (scRNA-seq) dataset composed of CD8+ T cells from mice challenged with acute and chronic Lymphocytic Choriomeningitis Virus (LCMV)(63). After recreating the published UMAP (Fig. 7F and Fig. S7), we performed ‘module score’ analysis to determine which regions (if any) are enriched for our core STAT5 signature and/or constituent gene sets (Table S1-S2). Regarding the latter, we found that all positive gene sets, whether derived from IL-7 or IL-15, naïve or memory CD8+ T cells, were enriched in a single region of the UMAP containing ‘early effector or exhausted’ cells (cluster 6; Fig. 7F and Fig. S7B). We interpret that this population had recently encountered antigen and, thus, were experiencing acute STAT5 activity downstream autocrine or paracrine IL-2 responses. By contrast, negative gene sets from naïve CD8+ T cells were enriched in different regions depending on whether driven by IL-7 or IL-15, while those derived from memory CD8+ T cells were not enriched at all (note scales; Fig. 7C). In most cases, narrowing to the top 50 elements based on fold change values enhanced module score performance, reflected in improved enrichment scores and consolidation of UMAP enrichment regions (Fig. S7B-C). Pan-cytokine gene sets from memory CD8+ T cells were far more enriched than those from naïve CD8+ T cells (Fig. S7D) and, crucially, our final core 85-element signature was highly and sharply enriched (Fig. 7F and S7A,D). Thus, our core signature and its constituents may be useful for detecting strong, synchronized STAT5 signaling associated with effector responses, but perhaps not weaker, asynchronous signaling associated with homeostatic responses.”

Reviewer 1 point-by-point (minor)

1. “Care should be taken to clearly state why each cytokine was used for each study (e.g. why IL-7 and IL-15 were used for RNA-seq studies, yet not IL-2)”

The following statement has been added to the Results in the revised manuscript:

“These cytokines were chosen because they are prominent in CD8+ T cell biology and mobilize STAT5 in both naïve and memory states (60).”

2. “The authors should consider including information regarding DEGs that were up- or down-regulated >2-fold or >3-fold; 1.5-fold is a relatively modest increase. This information may also help with core gene set identification, as DEG identification would be more stringent.”

The following statement has been added to the Results in the revised manuscript:

“A fold change threshold of >1.5-fold was chosen instead of the more popular >2-fold threshold to capture as many DEG as possible in scenarios where genotype effects were subtle.”

3. “For Figure 3D, an isotype control antibody stain should be included in each histogram plot. Further, repeated analyses should be enumerated, and statistical analysis should be performed in order for the stated conclusions to be adequately supported.”

Isotype controls are now included for all ‘total’ STAT5 cytometry. Pre-stimulus (no cytokine) controls are now included for all phospho-STAT5 cytometry. Neither isotope nor pre-stimulus controls are included for other makers as those studies involved highly vetted antibodies and met their primary goal of comparing expression across experimental groups.

Reviewer 2 point-by-point (minor)

1. Proofreading and figure errors

We thank Reviewer 2 for alerting us to proofreading and figure errors. We regret and corrected all of them.

2. “ the authors state that the WT/BBA chimera has “even more” reduction of splenic CD8+ T cells than the WT/AAB chimera. Whilst the selected plot shown in panel A reflects this strongly, statistics should be performed on the comparison of quantified ratios in panel C, as it does not really appear that there is a difference overall. The authors also state that the 1:5 ratio for the CD4+ T cells in the chimeras is “largely preserved”. While it is certainly closer than the CD8+ T cells, it is also reduced and the wording of the text should better reflect this.”

Following on the concern raised by Reviewer 2, we reappraised the bone marrow chimera studies and amended relevant text with more tempered conclusions (quoted below).

“Indeed, despite this consideration, splenic CD8+ T cell ratios strongly skewed towards WT donors, with the WT/BBA mix trending to greater shifts (Fig. 2A-C). By contrast, splenic CD4+ T cells had donor ratios closer to 1:1, suggesting that that they are less reliant on STAT5.”

3. “What do the dotted vertical lines in Figure 4B (left) represent? In the text it is stated that DEG called only in STAT5B deficient cells include targets such as Bcl2 and Myc, yet in the heatmap in Figure 4B (right) these and all other shown targets seem to be similarly dysregulated across both STAT5A and STAT5B deficient conditions. This is unclear.”

The dotted lines are drawn at values of -250 and 250 in Figure 4B to orient readers; few DEG sets go below or above these values. We regret that this was not effective due to lack of proper description in the legend, which we have corrected.

Regarding the heat map, we interpret that the bottom half is clearly darker than the top, reflecting greater effect of STAT5B deficiency. Given the scale on the right, we also interpret that many of the included genes were not called DEG in AAB cells (top). Most, including Bcl2 and Myc, are light pink colored which represents log2 fold change values of 0-0.5, below the 0.58 (1.5-fold) threshold for DEG calls.

4. “Figure 7A: the text states that here “STAT5B-driven transcriptomes” are compared across cytokines and differentiation states, but it is difficult to assess now which mouse genotypes are being assessed”

Text and legend have been amended in the revised manuscript to clarify that data from STAT5B-deficient cells was exclusively used to devise our STAT5 signatures.

5. “ It would be worth noting in the discussion that it has been previously shown that oncogenic (constitutively active) forms of STAT5 are much more promiscuous with respect to DNA binding and regulation of gene

expression at non-canonical loci than their endogenous 'normal' counterparts, which is likely why so many genes were differentially regulated in this system but not found in the integrated ChIP-seq dataset. They may not necessarily be indirect effects. “

As suggested by Reviewers 2 and 3, the revised manuscript now includes discussion of STAT5 mutations in T cell cancers, highlighting the fact that STAT5B mutations are both more common and more pathogenic than STAT5A mutations (quoted below).

Intro: “Also telling, somatic mutations of STAT5B are far more common in T cell cancers than STAT5A mutations and, when tested head-to-head, appear more pathogenic (38–40). Thus, functional asymmetry between STAT5 paralogs factors across human diseases and, as in mice, STAT5B appears dominant over STAT5A. However, it is important to note that, while less potent, gain-of-function STAT5A mutants are still capable of invoking T cell malignancy (38, 40, 41).”

Discussion: “Here, again, it is worth noting that GOF STAT5B mutations are more often associated with T cell cancers than STAT5A mutations and appear more potent in driving both cell intrinsic phenotypes and organismal level pathologies (38–40).”

We were unable to find publications showing that STAT5B mutants have an expanded target repertoire but do acknowledge the possibility that the constitutively active STAT5A construct that we used for retroviral studies could have atypically effects (quoted below).

Results: “Surprisingly, we found that most DEG were not engaged by STAT5A or STAT5B (Fig. 6B). We interpret this as evidence for indirect regulation (e.g. induction of secondary transcription factors, ‘piggybacking’ on other transcription factors), promiscuous binding due to supra-physiological expression (62) and/or atypical behavior of the constitutively active STAT5A construct used for these studies.”

Reviewer 3 point-by-point

1. “While the manuscript clearly highlights the importance of STAT5 signaling in CD8+ T cells, the review of the existing literature could be strengthened. In particular, more attention should be given to recent findings that connect STAT5 biology to cancer development and progression. Integrating these cancer-related insights would position the current study in a broader biological and translational framework. Doing so would also underscore the relevance of asymmetric paralog function not only in T cell biology but also in oncogenic signaling, thereby enhancing the manuscript’s impact for both immunology and cancer biology audiences.”

As suggested by Reviewers 2 and 3, the revised manuscript now includes discussion of STAT5 mutations in T cell cancers, highlighting the fact that STAT5B mutations are both more common and more pathogenic than STAT5A mutations (quoted below).

Intro: “Also telling, somatic mutations of STAT5B are far more common in T cell cancers than STAT5A mutations and, when tested head-to-head, appear more pathogenic (38–40). Thus, functional asymmetry between STAT5 paralogs factors across human diseases and, as in mice, STAT5B appears dominant over STAT5A. However, it is important to note that, while less potent, gain-of-function STAT5A mutants are still capable of invoking T cell malignancy (38, 40, 41).”

Discussion: “Here, again, it is worth noting that GOF STAT5B mutations are more often associated with T cell cancers than STAT5A mutations and appear more potent in driving both cell intrinsic phenotypes and organismal level pathologies (38–40).”

2. “...given the translational emphasis, a discussion of whether these findings are conserved in human cells would further strengthen the manuscript and broaden its impact.”

As suggested by Reviewers 3, we added human transcript data from ImmGen and DICE comparing levels of *STAT5A* and *STAT5B* in naive and effector/memory T cells. Associated text is quoted below:

Results: “Similar disparity was also evident at transcriptional level where, again, Stat5b accounts for 60-70% of total Stat5 mRNA (Fig. 3F) and holds true whether transcriptomes were generated in-house (detailed below) or mined from public databases (Fig. 3F and Fig. S3A-C), and whether sourced from mouse or human T cells (Fig. S3A,D).”